# No More Pesky Hyperparameters: Offline Hyperparameter Tuning for RL

**Han Wang**[*][†]                                                              *han8@ualberta.ca*

**Archit Sakhadeo**[*][†]                                                       *sakhadeo@ualberta.ca*

**Adam White**[†]                                                              *amw8@ualberta.ca*

**James Bell**[†]                                                              *jbell1@ualberta.ca*

**Vincent Liu**[†]                                                             *vliu1@ualberta.ca*

**Xutong Zhao**[†]                                                             *xutong@ualberta.ca*

**Puer Liu**[†]                                                                *puer@ualberta.ca*

**Tadashi Kozuno**[†]                                                          *kozuno@ualberta.ca*

**Alona Fyshe**[†]                                                             *alona@ualberta.ca*

**Martha White**[†]                                                            *whitem@ualberta.ca*

**Reviewed on OpenReview:** *https://openreview.net/forum?id=AiOUi3440V*

## Abstract

The performance of reinforcement learning (RL) agents is sensitive to the choice of hyperparameters. In real-world settings like robotics or industrial control systems, however, testing different hyperparameter configurations directly on the environment can be financially prohibitive, dangerous, or time consuming. We focus on hyperparameter tuning from offline logs of data, to fully specify the hyperparameters for an RL agent that learns online in the real world. The approach is conceptually simple: we first learn a model of the environment from the offline data, which we call a calibration model, and then simulate learning in the calibration model to identify promising hyperparameters. Though such a natural idea is (likely) being used in industry, it has yet to be systematically investigated. We identify several criteria to make this strategy effective, and develop an approach that satisfies these criteria. We empirically investigate the method in a variety of settings to identify when it is effective and when it fails.

## 1 Introduction

Reinforcement learning (RL) agents are extremely sensitive to the choice of hyperparameters that regulate speed of learning, exploration, degree of bootstrapping, amount of replay and so on. The vast majority of work RL is focused on new algorithmic ideas and improving performance—in both cases assuming near-optimal hyperparameters. Indeed the vast majority of empirical comparisons involve well-tuned implementations or reporting the best performance after a hyperparameter sweep. Although progress has been made towards eliminating the need for tuning with adaptive methods (White & White, 2016; Xu et al., 2018; Mann et al.,

---

[*]These authors contributed equally to this work.

[†]Computing Science, Alberta Machine Intelligence Institute (Amii), University of Alberta, Edmonton, Alberta, Canada.

2016; Zahavy et al., 2020; Jacobsen et al., 2019; Kingma & Ba, 2014; Papini et al., 2019), widely used agents employ dozens of hyperparameters and tuning is critical to their success (Henderson et al., 2018).

The reason domain specialization and hyperparameter sweeps are possible—and perhaps why our algorithms are so dependent on them—is because most empirical work in RL is conducted in simulation. Simulators are critical for research because they facilitate rapid prototyping of ideas and extensive analysis. On the other hand, simulators allow us to rely on features of simulation not possible in the real world, such as exhaustively sweeping different hyperparameters. Often, it is not acceptable to test poor hyperparameters on a real system that could cause serious failures. In many cases, interaction with the real system is limited, or in more extreme cases, only data collected from a human operator is available. Recent experiments confirm significant sensitivity to hyperparameters is exhibited on real robots as well (Mahmood et al., 2018). It is not surprising that one of the major roadblocks to applied RL is extreme hyperparameter sensitivity.

Fortunately, there is an alternative to evaluating algorithms on the real system: using previously logged data under an existing controller (human or otherwise). This offline data provides some information about the system, which could be used to evaluate and select hyperparameters without interacting with the real world. Hyperparameters are general, and can even include a policy initialization that is adjusted online. We call this problem *Data2Online*.

This problem setting differs from the standard offline or batch RL setting because the goal is to select *hyperparameters* offline for the agent to use to *learn online in deployment*, as opposed to learning a *policy* offline. Typically in offline RL a policy is learned on the batch of data, using a method like Fitted Q Iteration (FQI) (Ernst et al., 2005; Riedmiller, 2005; massoud Farahmand et al., 2009), and the resulting fixed policy is deployed without being updated. Our setting is less stringent, as the policy continually adapts during deployment. Intuitively, selecting just the hyperparameters for further online learning should not suffer from the same hardness problems as offline RL (see (Wang et al., 2021) for hardness results), because the agent has the opportunity to gather more data online and adjust its policy.

We discuss a strategy to use offline data for selecting hyperparameters for learning from scratch in deployment. The idea is simple: we use the data to learn a *calibration model*, and evaluate hyperparameters in the calibration model. For example, consider designing a learning system for controlling a water treatment plant, given only a set of data logs, visualized in Figure 1. We want an agent to control pump speeds and chemical treatments to clean the water with minimal energy usage—but how do we set the learning rate and other hyperparameters of this agent? We can learn a calibration model offline from data logs previously collected while human operators controlled the plant. The agent can be tested with different hyperparameter settings in the calibration model, to gauge which resulted in best online performance (e.g., reward accumulation or stable learning). The agent with these best hyperparameters is then deployed on the real plant.

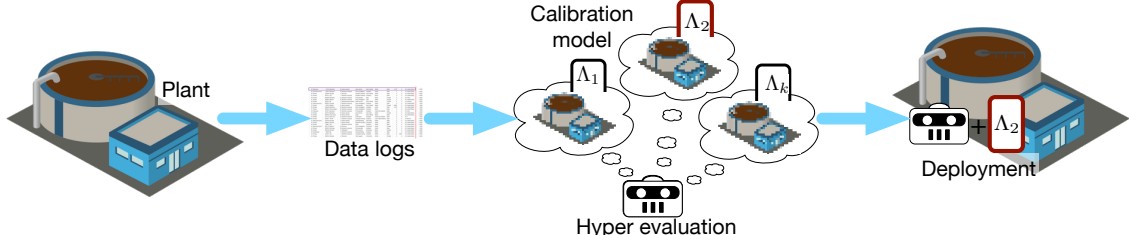

Figure 1: Data2Online: each hyperparameter setting is denoted by Λ. The plant imagined by the agent (via calibration model) is intentionally pixilated to reflect approximation of the true plant.

This idea is natural, and versions of it are likely being used in industry already. For example, one could first construct *construct* a high-fidelity simulator and then use RL. This strategy was taken for impressive applications like for flying balloons (Bellemare et al., 2020) and for controlling plasma (Degrave et al., 2022). Still other work has *learned* slightly less high-fidelity simulators, using (recurrent) neural networks, to prototype and compare algorithms, such as for gas turbine control (Schaefer et al., 2007). These are natural steps, to understand which algorithms are effective, before deploying on a real system!

The primary novelty in our work is to investigate learning and using calibration models as a black-box algorithm for Data2Online, particularly in settings where we do not have access to a physics simulator of the deployment scenario. Instead of considering the use of a simulator as part of the engineer's prototyping, we consider the learning and use of a calibration model as part of an automated process. The input to the automation pipeline is logged data from interaction under the current (human) operator and the output is an agent that learns in deployment (not a static policy). It may seem like a hopeless endeavour, considering the evidence in Sim2Real and in model-based RL suggesting small errors in models can result in poor policies (Talvitie, 2017; Höfer et al., 2021). However, the key distinction for our setting is that we identify *hyperparameters*, rather than learning a complete policy. The calibration model need not be a perfect simulator to be useful for identifying reasonable hyperparameters, whereas learning a transferable policy likely requires accurate models.

This automated process is what we call Data2Online, and there has as yet been very little work understanding theoretical limits, algorithms and potential failure modes. To the best of our knowledge, there is only one other work that considers how to use a fixed offline dataset to evaluate an agent that learns online in deployment (Mandel et al., 2016). Their approach uses a novel strategy to reason about how the policy changes during learning, but is only effective for short horizon problems.

In this paper, we first introduce our Data2Online strategy, and outline conditions on the calibration model and learning agents for this strategy to be effective. We bound the difference in the value in the real environment of the hyperparameters chosen in the calibration model, to the true best hyperparameters, in terms of the calibration model error and length of interaction. We then develop a non-parametric calibration model, based on k-nearest neighbors and a carefully chosen distance metric, that satisfies the conceptual criteria needed for the calibration model. We investigate the approach in three problems with different types of learning dynamics, two different learning agents, under different offline data collection policies, and with ablations on the key components of our proposed calibration model. We conclude by highlighting that grid search can be replaced with any hyperparameter optimization algorithm, and that this further improves performance in the real environment.

## 2 Related Problem Settings

Offline RL involves learning from a dataset, but with a different goal: to deploy a fixed (near-optimal) policy. As a result, the hyperparameter selection approaches for offline RL are quite different. One strategy that has been used is to evaluate different policies corresponding to different hyperparameter settings, under what has been introduced as the Offline Policy Selection problem (Yang et al., 2020). This setting differs from our setting in that they evaluate the utility of the learned policy, rather than the utility of hyperparameters for learning online, in deployment. Some work in offline RL examines learning from data, and then adapting the policy online (Ajay et al., 2021; Yang & Nachum, 2021; Lee et al., 2021), including work that alternates between data collection and high confidence policy evaluation (Chandak et al., 2020a;b). Our problem is complementary to these, as a strategy is needed to select their hyperparameters.

In Sim2Real the objective is to construct a high fidelity simulator of the deployment setting, and then transfer the policy and, in some cases, continue to fine tune in deployment. We focus on learning the calibration model from collected data, whereas in Sim2Real the main activity is designing and iterating on the simulator itself (Peng et al., 2018). Again, however, approaches for Data2Online are complementary, and even provide another avenue to benefit from the simulator developed in Sim2Real to pick hyperparameters for fine tuning.

Domain adaptation in RL involves learning on a set of source tasks, to transfer to a target task. The most common goal has been to enable zero-shot transfer, where the learned policy is fixed and deployed in the target task (Higgins et al., 2017; Xing et al., 2021). Our problem has some similarity to domain adaptation, in that we can think of the calibration model as the source task and the real environment as the target task. Domain adaptation, however, is importantly different than our Data2Online problem: (a) in our setting we train in a *learned* calibration model not a real environment, and need a mechanism to learn that model (b) the relationship between our source and target is different than the typical relationship in domain adaptation and (c) our goal is to select and transfer hyperparameters, not learn and transfer policies.

Learning from demonstration (LfD) and imitation learning involve attempting to mimic or extract a policy at least as good as a demonstrator. If the agent is learning to imitate online, then it is unrealistic to assume the demonstrator would generate enough training data required to facilitate hyperparameter sweeps. If the learner's objective is to imitate from a dataset, then this is exactly the problem study of this paper. Unfortunately, hyperparameter tuning in LfD is usually not addressed; instead it is common to use explicit sweeps (Merel et al., 2017; Barde et al., 2020; Ghasemipour et al., 2019; Behbahani et al., 2019) or manual, task-specific tuning (Finn et al., 2017). Hyperparameter tuning, however, is a major obstacle to the deployment of LfD methods (Ravichandar et al., 2020).

Finally, there is a large literature on hyperparameter selection in RL. Most introduce meta algorithms that learn hyperparameters, including work on meta-descent for stepsizes (Sutton, 1992; Xu et al., 2018; Jacobsen et al., 2019) and selecting the trace parameter (Downey & Sanner, 2010; Mann et al., 2016; White & White, 2016). These algorithms could be beneficial for offline hyperparameter selection, because they help reduce sensitivity to hyperparameters; but they are not a complete solution as they still have hyperparameters to tune. Other work has provided parameter-free methods that have theoretically defined formulas for hyperparameters (Papini et al., 2019). Deriving such algorithms is important, but is typically algorithm specific and requires time to extend to broader classes of algorithms. Finally, recent work has examined online hyperparameter selection, using off-policy learning to assess the utility of different hyperparameters in parallel (Paul et al., 2019; Tang & Choromanski, 2020). Otherwise, much of the work has been focused on settings where it is feasible to obtain multiple runs under different hyperparameters—such as in simulation— with the goal to improve on simple grid search (Srinivas et al., 2010; Bergstra & Bengio, 2012; Snoek et al., 2012; Li et al., 2018; Jaderberg et al., 2017; Falkner et al., 2018; Parker-Holder et al., 2020).

## 3 Problem Formulation

In RL, an agent learns to make decisions through interaction with an environment. We formulate the problem as a Markov Decision Process (MDP), described by the tuple $(\mathcal{S}, \mathcal{A}, \mathcal{R}, \mathcal{P})$. $\mathcal{S}$ is the state space and $\mathcal{A}$ the action space. $\mathcal{R} : \mathcal{S} \times \mathcal{A} \times \mathcal{S} \to \mathbb{R}$ is the reward, a scalar returned by the environment. The transition probability $\mathcal{P} : \mathcal{S} \times \mathcal{A} \times \mathcal{S} \to [0,1]$ describes the probability of transitioning to a state, for a given state and action. On each discrete timestep $t$ the agent selects an action $A_t$ in state $S_t$, the environment transitions to a new state $S_{t+1}$ and emits a scalar reward $R_{t+1}$.

The agent's objective is to find a policy that maximizes future reward. A policy $\pi : S \times A \to [0,1]$ defines how the agent chooses actions in each state. The objective is to maximize future discounted reward or the *return*, $G_t \doteq R_{t+1} + \gamma_{t+1} G_{t+1}$ for a discount $\gamma_{t+1} \in [0,1]$ that depends on the transition $(S_t, A_t, S_{t+1})$ (White, 2017). For continuing problems, the discount may simply be a constant less than 1. For episodic problems the discount might be 1 during the episode, and become zero when $S_t, A_t$ leads to termination. Common approaches to learn such a policy are Q-learning and Expected Sarsa, which approximate the action-values— the expected return from a given state and action—and Actor-Critic methods that learn a parameterized policy (see (Sutton & Barto, 2018)).

We assume that in the *Data2Online* setting the agent has access to an offline log of data that it can use to initialize hyperparameters before learning online. This log consists of $n_{\text{data}}$ tuples of experience $\mathcal{D} = \{(S_t, A_t, R_{t+1}, S_{t+1}, \gamma_{t+1})\}_{i=1}^{n_{\text{data}}}$, generated by interaction in the environment by a previous controller or controllers.[1] For example, an agent that will use Expected Sarsa might want to use this data to decide on a suitable stepsize $\alpha$, the number of layers $l$ in the neural network (NN) architecture for the action-values and even an initialization $\theta_0$ for the NN parameters—namely a policy initialization. There are several options for each hyperparameter combination, $\lambda = (\alpha, l, \theta_0)$, resulting in a set of possible hyperparameters $\Lambda$ to consider. This set can be discrete or continuous, depending on the underlying ranges. For example, the agent might want to consider any $\alpha \in [0,1]$ and a $\theta_0$ only from a set of three possible choices.

Procedurally, the Data2Online algorithm is given the dataset $\mathcal{D}$ and the set of hyperparameters $\Lambda$, and outputs a selected hyperparameter setting $\tilde{\lambda}$. A good choice is one that is within $\varepsilon$-optimal of the best

---

[1]Going back to our water treatment example, this data might have been collected over the last two years using actions from a human operator. We expect the data is likely from a reasonable or nearly optimal policy. Before letting our agent learn on and control the water treatment plant, it is sensible to use this already available data to improve its performance online.

hyperparameters

$$\text{Perf}(\tilde{\lambda}) \geq \max_{\lambda \in \Lambda} \text{Perf}(\lambda) - \varepsilon \tag{1}$$

where $\text{Perf}(\lambda)$ is the online performance of the agent, when deployed with the given hyperparameters. Typically, this will be the cumulative reward for continuing problems and average return for episodic problems, for a fixed number of steps $T$. Many hyperparameters may allow the agent to perform well, so we focus on nearly-optimal performance under $\tilde{\lambda}$ rather than on identifying the best hyperparameters.

The central question for this Data2Online problem is: how can the agent use this log of data to select hyperparameters before learning in deployment? This is no easy task. The agent cannot query $\text{Perf}(\lambda)$. It is not only evaluating a *fixed* policy, for which we could use the large literature on Off-policy Policy Evaluation. It is evaluating a *learning* policy. In the remainder of this paper, we introduce and test a new algorithm for this Data2Online problem.

## 4    Data2Online using Calibration Models

This section introduces the idea of calibration models and how they can be used for hyperparameter selection. We first discuss how to use the calibration model to select hyperparameters, before providing a way to learn the calibration model. We then discuss certain criteria on the calibration model and agent algorithm that make this strategy more appropriate. We conclude with some theoretical characterization of the error in hyperparameter selection, based on inaccuracies in the calibration model.

### 4.1    Using Calibration Models to Select Hyperparameters

A calibration model is a simulator of the environment—learned from an offline batch of data—used to specify (or calibrate) hyperparameters in the agent. With the calibration model, the agent can test different hyperparameter settings. It evaluates the online performance of different hyperparameter choices in the calibration model and selects the best one. It can then be deployed into the true environment without any remaining hyperparameters to tune.

The basic strategy is simple: we train a calibration model, then do grid search in the calibration model and pick the top hyperparameter, as summarized in Algorithm 1. For each hyperparameter, we obtain a measure of the online performance of the agent across $n_{\text{runs}}$ in the calibration model, assuming it gets to learn for $n_{\text{steps}}$ of interaction. The pseudocode for AgentPerfInEnv is in Algorithm 2, for the episodic setting where we report average returns during learning. Note that we add a cutoff in the evaluation scheme to guarantee that at least 30 episodes are seen by the agent during training in the calibration model. We cut off episodes that run longer than $n_{\text{steps}}/30$ steps, teleporting the agent back to a start state.

---

**Algorithm 1** Hyperparameter Selection with Calibration Models using Grid Search

---

**Input:** $\boldsymbol{\Lambda}$: hyperparameter set for learner *Agent*
$\mathcal{D}$: the offline data log
$n_{\text{steps}}$: number of interactions or steps
$n_{\text{runs}}$: number of runs
   Train calibration model $\mathcal{C}$ with $\mathcal{D}$
  **for** $\lambda$ in $\boldsymbol{\Lambda}$ **do**
    $\text{Perf}[\lambda] = \text{AgentPerfInEnv}(\mathcal{C}, \textit{Agent}(\lambda), n_{\text{steps}}, n_{\text{runs}})$    (see Algorithm 2)
  **end for**
**Return:** $\arg\max_{\lambda \in \boldsymbol{\Lambda}} \text{Perf}[\lambda]$

---

Many components in this approach are modular, and can be swapped with other choices. For example, instead of expected return during learning (online performance), optimizing the hyperparameters might be more desirable to find the best policy after a budget of steps. This would make sense if cumulative reward during learning in deployment was not important. We might also want a more robust agent, and instead of expected return, we may want median return. Finally, the grid search can be replaced with a more efficient hyperparameter selection method; we discuss this further in Section 7.

---

**Algorithm 2** AgentPerfInEnv

---

**Input:** $\mathcal{C}$: calibration model, *Agent*: learner, $n_{\text{steps}}$: # of steps, $n_{\text{runs}}$: # of runs

   $n_{\text{cutoff}} \leftarrow n_{\text{steps}}/30$    ▷ Ensure there are at least 30 episodes
   $ReturnsAcrossRuns = []$
   **for** $i = 1 \ldots n_{\text{runs}}$ **do**
      $t \leftarrow 0, G \leftarrow 0, Returns = []$
      $s \leftarrow$ random start state from $\mathcal{C}$
      **for** $j = 1 \ldots n_{\text{steps}}$ **do**
         Obtain $a$ from $Agent(s)$, obtain $s', r = \mathcal{C}(s, a)$, give $s', r$ to $Agent$
         $G \leftarrow G + \gamma^t r$
         $t \leftarrow t + 1$
         **if** $s'$ is terminal or $t > n_{\text{cutoff}}$ **then**
            Append $G$ to $Returns$
            $s \leftarrow$ random start state from $\mathcal{C}$, $t \leftarrow 0, G \leftarrow 0$
         **end**
      **end for**
      $ReturnsAcrossRuns[i] \leftarrow average(Returns)$
   **end for**
**Return:** $average(ReturnsAcrossRuns)$

---

We can also make this hyperparameter search more robust to error in the calibration model by obtaining performance across an ensemble of calibration models. This involves using $n_{\text{ensembles}}$ random subsets of the log data, say by dropping at random 10% of samples, and training $n_{\text{ensembles}}$ calibration models. The hyperparameter performance can either be averaged across these models, or a more risk-averse criterion could be used like worst-case performance. Using an average across models is like using a set of source environments to select hyperparameters—rather than a single source—and so could improve transfer to the real environment.

These are all additions to this new Data2Online strategy. The central idea is to use this calibration model to evaluate hyperparameters, as if we had a simulator. We, therefore, investigate this idea first in its simplest form with expected returns, grid search and only one calibration model.

### 4.2 When is this Approach Effective?

This section highlights three conceptual criteria for designing the calibration model and selecting agents for which Data2Online should be effective. This includes 1) stability under model iteration, 2) handling actions with low data coverage and 3) selecting agent algorithms that only have initialization hyperparameters, namely those that affect early learning but diminish in importance over time.

Producing reasonable transitions under many steps of model iteration is key for the calibration model. The calibration model is iterated for many steps, because the agent interacts with the calibration model as if it were the true environment—for an entire learning trajectory. An agent often relies on random exploration to discover good rewards before learning can adapt the policy. Thus the first few episodes may be extremely long before a goal state is found. In Mountain Car, for example, a linear function approximation agent takes over 65,000 steps to wander to the goal, and a nonlinear function approximator may take even longer.[2]

It is key, therefore, that the calibration model be *stable* and *self-correcting*. A stable model is one where, starting from any initial state in a region, the model remains in that region. A self-correcting model is one that, even if it produces a few non-real states, it comes back to the space of real states. Otherwise,

---

[2]It might seem like a solution is to have short episode cutoffs. However, some deployment scenarios may not allow short cutoffs, and we want our calibration model to reflect the real-world as best we can. Further, in continuing problems there are no terminations at all. Finally, cutting off episodes typically makes the exploration problem much easier and thus may indeed impact the hyperparameters selected.

model iteration can produce increasingly non-sensical states, as has been repeatedly shown in model-based RL (Talvitie, 2017; Jafferjee et al., 2020; Abbas et al., 2020; Chelu et al., 2020).

The model also needs to handle actions with no coverage, or low coverage. For unvisited or unknown states, the model simply does not include such states. The actions, however, can be queried from each state. If an action has not been taken in a state, nor a similar state, the model cannot produce a reasonable transition. Any choice will have limitations, because inherently we are filling in this data gap with an inductive bias. A common choice in offline RL is to assume pessimistic transitions: if an action is not observed, it is assumed to have a bad outcome. This ensures the learned, fixed policy avoids these unknown areas.

The choice is even more nuanced in Data2Online. Just like offline RL, it can be sensible to avoid these unknown actions, to answer: in the space known to the agent, what hyperparameters allow the agent to learn quickly? But, another plausible alternative is that we want to select hyperparameters to encourage the agent to explore unknown areas, since once deployed the agent can update its policy in these unknown areas. In other words, a principle of optimism could also be sensible. Selecting the right inductive bias will depend on the environment and the types of hyperparameters we are selecting. This question will likely be one of the largest questions in Data2Online, similarly to how it remains an open question in offline RL.

The third criterion is a condition on the agent, rather than the model. Practically, we can only test each hyperparameter setting for a limited number of steps in the calibration model. So, the calibration model is only simulating early learning. This suggests that this approach will be most effective if we tune *initialization hyperparameters*: those that provide an initial value for a constant but wash away over time. Examples include an initial learning rate which is then adapted; policy initialization; and an initial architecture that is grown and pruned over time.

These criteria are conceptual, based on reasoning through when we expect success or failure. We use these conceptual criteria to propose an appropriate approach to learn a calibration model in the next section. In addition to conceptual reasoning, theoretical understanding of the Data2Online problem setting is also critical. We provide a first step in that direction in the next section.

### 4.3 Theoretical Insights

This problem has aspects that both make it harder and potentially easier than a standard offline RL problem. One aspect that makes this problem harder is that we have to evaluate a learning policy offline, rather than a fixed learned one. A fixed policy can be assessed using policy evaluation, and there exists a variety of theoretical results on the quality of those estimates, many based on the foundational simulation lemma (Kearns & Singh, 2002). No such results exist for evaluating a policy that will be changing online.

At the same time, intuitively, the problem could be easier than the offline RL problem, because the policy adapts online. Instead, we only have to select from a potentially small number of hyperparameters, rather than from a potentially large policy space. For example, it may be relatively easy to identify the best stepsize out of a set of three stepsizes. Further, if a policy learning algorithm is chosen that is robust to its hyperparameter settings, then the problem may be even simpler. For example, it may be simple to select the initial stepsize for an adaptive stepsize algorithm, where the initial stepsize only influences the magnitude of updates for the first few steps.

We first extend the foundational simulation lemma to the Data2Online setting, in Theorem 1. Then, in Theorem 2, we show how to use this result, to bound how far the value of the hyperparameters chosen in the learned calibration model are from the best hyperparameters. Finally, we discuss how it might be possible to formalize this second intuition, for future theoretical investigation.

We start by defining some needed terms. An online learner can be characterized by a history dependent policy (see Chapter 38 of (Lattimore & Szepesvári, 2020)). A history dependent policy is $\pi = (\pi_0, \pi_1, \pi_2, \dots)$ where $\pi_t : \mathcal{H}_t \to \Delta(\mathcal{A})$ and $\mathcal{H}_t = (\mathcal{S} \times \mathcal{A} \times \mathbb{R})^t \times \mathcal{S}$ is the history at time step $t$. For simplicity, we assume the rewards are deterministic in $[0, r_{\max}]$ and the MDP has one initial state $s_0$. The online learning agent interacts with the environment for $T$ steps in total, in either a continuing or fixed-horizon setting.

The value function for this online learner $\pi$ is the sum of rewards from time $t$ to the end of learning at $T-1$

$$V_t^\pi(h_t) = \mathbb{E}\left[\sum_{t'=t}^{T-1} r(S_{t'}, A_{t'}) \mid H_t = h_t\right]$$

where the expectation is under $A_t \sim \pi_t(\cdot \mid H_t)$, $S_{t+1} \sim P(\cdot \mid S_t, A_t)$. Note that $V_0^\pi(s_0)$ is the $T$ step objective that we use to select hyperparameters. For the fixed-horizon setting where episodes are of length *horizon*, $T = K \cdot horizon$ where $K$ is the number of episodes. In this setting, the expectation is under $S_{t+1} \sim P(\cdot \mid S_t, A_t)$ if $t$ is not the last step of an episode and $S_{t+1} = s_0$ if $t$ is the last step of an episode. Dividing $V_0^\pi(s_0)$ by $K$ gives the average episodic return over $K$ episodes.

**Theorem 1** (Simulation Lemma for Online Learners). *Assume the rewards $r(s,a)$ are deterministic in $[0, r_{max}]$ and the MDP has one initial state $s_0$ Suppose we have a learned model $(\hat{P}, \hat{r})$ such that*

$$\|\hat{P}(\cdot \mid s,a) - P(\cdot \mid s,a)\|_1 \le \varepsilon_p \quad and \quad |r(s,a) - \hat{r}(s,a)| \le \varepsilon_r \qquad for\ all\ (s,a) \in \mathcal{S} \times \mathcal{A}$$

*and that $\hat{r}(s,a) \in [0, r_{max}]$. Let $\hat{V}_t^\pi(\cdot)$ denote the value function under the learned model. Then for any history dependent policy $\pi$, we have that*

$$|V_0^\pi(s_0) - \hat{V}_0^\pi(s_0)| \le \varepsilon_r T + \frac{\varepsilon_p r_{max} T^2}{2}.$$

*Proof.* We follow the proof of the simulation lemma. Since the rewards are deterministic, the history does not need to contain reward, that is, $\mathcal{H}_t = (\mathcal{S} \times \mathcal{A})^t \times \mathcal{S}$. For any $h_t = (s_0, a_t, \ldots, s_t) \in \mathcal{H}_t$ with $t < T-1$,

$$V_t^\pi(h_t) = \sum_{a \in \mathcal{A}} \pi_t(a \mid h_t) r(s_t, a) + \sum_{a \in \mathcal{A}} \pi_t(a \mid h_t) \sum_{s' \in \mathcal{S}} P(s, a, s') V_{t+1}^\pi((h_t, a, s'))$$

$$\hat{V}_t^\pi(h_t) = \sum_{a \in \mathcal{A}} \pi_t(a \mid h_t) \hat{r}(s_t, a) + \sum_{a \in \mathcal{A}} \pi_t(a \mid h_t) \sum_{s' \in \mathcal{S}} \hat{P}(s, a, s') \hat{V}_{t+1}^\pi((h_t, a, s'))$$

and for the last step we have

$$V_{T-1}^\pi(h_{T-1}) = \sum_{a \in \mathcal{A}} \pi_{T-1}(a \mid h_{T-1}) r(s_{T-1}, a) \quad and \quad \hat{V}_{T-1}^\pi(h_{T-1}) = \sum_{a \in \mathcal{A}} \pi_{T-1}(a \mid h_{T-1}) \hat{r}(s_{T-1}, a).$$

We prove the simulation lemma from the last step. For $t = T-1$,

$$\left|V_{T-1}^\pi(h_{T-1}) - \hat{V}_{T-1}^\pi(h_{T-1})\right| = \left|\sum_{a \in \mathcal{A}} \pi_{T-1}(a \mid h_{T-1}) r(s_{T-1}, a) - \sum_{a \in \mathcal{A}} \pi_{T-1}(a \mid h_{T-1}) \hat{r}(s_{T-1}, a)\right| \le \varepsilon_r.$$

For all $t < T-1$,

$$|V_t^\pi(h_t) - \hat{V}_t^\pi(h_t)| = |\sum_{a \in \mathcal{A}} \pi_t(a \mid h_t) r(s_t, a) + \sum_{a \in \mathcal{A}} \pi_t(a \mid h_t) \sum_{s' \in \mathcal{S}} P(s, a, s') V_{t+1}^\pi((h_t, a, s'))$$

$$- \sum_{a \in \mathcal{A}} \pi_t(a \mid h_t) \hat{r}(s_t, a) - \sum_{a \in \mathcal{A}} \pi_t(a \mid h_t) \sum_{s' \in \mathcal{S}} \hat{P}(s, a, s') \hat{V}_{t+1}^\pi((h_t, a, s'))|$$

$$\le \varepsilon_r + \sum_{a \in \mathcal{A}} \pi_t(a \mid h_t) |\sum_{s' \in \mathcal{S}} P(s, a, s') V_{t+1}^\pi((h_t, a, s')) - \sum_{s' \in \mathcal{S}} \hat{P}(s, a, s') \hat{V}_{t+1}^\pi((h_t, a, s'))|$$

$$= \varepsilon_r + \sum_{a \in \mathcal{A}} \pi_t(a \mid h_t) |\sum_{s' \in \mathcal{S}} P(s, a, s') V_{t+1}^\pi((h_t, a, s')) - \sum_{s' \in \mathcal{S}} \hat{P}(s, a, s') V_{t+1}^\pi((h_t, a, s'))$$

$$+ \sum_{s' \in \mathcal{S}} \hat{P}(s, a, s') V_{t+1}^\pi((h_t, a, s')) - \sum_{s' \in \mathcal{S}} \hat{P}(s, a, s') \hat{V}_{t+1}^\pi((h_t, a, s'))|$$

$$\le \varepsilon_r + \sum_{a \in \mathcal{A}} \pi_t(a \mid h_t) |\sum_{s' \in \mathcal{S}} (P(s, a, s') - \hat{P}(s, a, s')) \underbrace{V_{t+1}^\pi((h_t, a, s'))}_{\le (T-t-1) r_{\max}}|$$

$$+ \sum_{a \in \mathcal{A}} \pi_t(a \mid h_t) |\sum_{s' \in \mathcal{S}} \hat{P}(s, a, s') (V_{t+1}^\pi((h_t, a, s')) - \hat{V}_{t+1}^\pi((h_t, a, s')))|$$

$$\leq \varepsilon_r + \sum_{a \in \mathcal{A}} \pi_t(a \mid h_t)(T - t - 1)r_{\max} \sum_{s' \in \mathcal{S}} \mid P(s, a, s') - \hat{P}(s, a, s') \mid + \max_{a, s'} |V_{t+1}^{\pi}((h_t, a, s')) - \hat{V}_{t+1}^{\pi}((h_t, a, s'))|$$

$$\leq \varepsilon_r + \varepsilon_p r_{\max}(T - t - 1) + \max_{a, s'} |V_{t+1}^{\pi}((h_t, a, s')) - \hat{V}_{t+1}^{\pi}((h_t, a, s'))|$$

Therefore, $|V_0^{\pi}(s_0) - \hat{V}_0^{\pi}(s_0)| \leq \varepsilon_r + \varepsilon_p r_{\max}(T - 1) + \cdots + \varepsilon_r + \varepsilon_p r_{\max} 1 + \varepsilon_r \leq \varepsilon_r T + \frac{\varepsilon_p r_{\max} T^2}{2}.$ $\qquad \square$

Theorem 1 says that if we have a good model of the environment, we can evaluate the $T$ step objective for any online learner with bounded error. In particular, we can control this evaluation error by controlling the error of the learned model. Note that $v_{\max} \doteq T r_{\max}$ is the maximum value, so the last term $\frac{\varepsilon_p r_{\max} T^2}{2}$ can be interpreted as $\frac{\varepsilon_p v_{\max} T}{2}$, meaning the bound scales with $T$: $(\varepsilon_r + \frac{\varepsilon_p v_{\max}}{2})T$.

Back to our problem setting. Let $\Lambda$ be the set of hyperparameters and $\pi_\lambda$ be a learner's policy with $\lambda \in \Lambda$. In our algorithm, we choose the best hyperparameters based on $\tilde{V}_0^{\pi}(s_0)$, which is an estimator for $\hat{V}_0^{\pi}(s_0)$ by running $n$ runs with $\hat{P}$. Let $\tilde{\lambda} = \arg\max_{\lambda \in \Lambda} \tilde{V}_0^{\pi_\lambda}(s_0)$ be the hyperparameters returned by our algorithm and $\lambda^* = \arg\max_{\lambda \in \Lambda} V_0^{\pi_\lambda}(s_0)$ be the true best hyperparameters in the set. The following theorem shows that our hyperparameters will not be too far from the best hyperparameters in terms of the $T$ step objective.

**Theorem 2.** *Under the same conditions as Theorem 1, with probability $1 - \delta$, we have*

$$V_0^{\pi_{\lambda^*}}(s_0) - V_0^{\pi_{\tilde{\lambda}}}(s_0) \leq 2\varepsilon_r T + \varepsilon_p r_{max} T^2 + r_{max} T \sqrt{\frac{2 \ln(4/\delta)}{n}}$$

$$= \underbrace{(2\varepsilon_r + \varepsilon_p v_{max})T}_{approximation\ error} + \underbrace{v_{max}\sqrt{\frac{2 \ln(4/\delta)}{n}}}_{estimation\ error}.$$

*Proof.* By Hoeffding's inequality, for a given $\pi$, we have with probability $1 - \delta/2$ that

$$|\hat{V}_0^{\pi}(s_0) - \tilde{V}_0^{\pi}(s_0)| \leq T r_{\max} \sqrt{\frac{\ln(4/\delta)}{2n}}$$

because the return in each run, to give the sample average $\tilde{V}_0^{\pi}(s_0)$, is in $[0, T r_{\max}]$. Using the union bound, we can say this inequality holds for both $\pi_{\lambda^*}$ and $\pi_{\tilde{\lambda}}$, with probability $1 - \delta$. The source of this difference is from using a limited number of runs to approximate $\hat{V}_0^{\pi}(s_0)$. As we increase the number of runs $n$, then the difference between our estimator $\tilde{V}_0^{\pi}(s_0)$ and $\hat{V}_0^{\pi}(s_0)$ goes to zero.

Now we can reason about the hyperparameters chosen using $\tilde{\lambda} = \arg\max_{\lambda \in \Lambda} \tilde{V}_0^{\pi_\lambda}(s_0)$.

$$\begin{aligned}
V_0^{\pi_{\lambda^*}}(s_0) - V_0^{\pi_{\tilde{\lambda}}}(s_0) &= V_0^{\pi_{\lambda^*}}(s_0) - \tilde{V}_0^{\pi_{\tilde{\lambda}}}(s_0) + \tilde{V}_0^{\pi_{\tilde{\lambda}}}(s_0) - V_0^{\pi_{\tilde{\lambda}}}(s_0) \\
&\leq V_0^{\pi_{\lambda^*}}(s_0) - \tilde{V}_0^{\pi_{\lambda^*}}(s_0) + \tilde{V}_0^{\pi_{\tilde{\lambda}}}(s_0) - V_0^{\pi_{\tilde{\lambda}}}(s_0) \\
&= V_0^{\pi_{\lambda^*}}(s_0) - \tilde{V}_0^{\pi_{\lambda^*}}(s_0) + \hat{V}_0^{\pi_{\lambda^*}}(s_0) - \hat{V}_0^{\pi_{\lambda^*}}(s_0) \\
&\quad + \tilde{V}_0^{\pi_{\tilde{\lambda}}}(s_0) - V_0^{\pi_{\tilde{\lambda}}}(s_0) + \hat{V}_0^{\pi_{\tilde{\lambda}}}(s_0) - \hat{V}_0^{\pi_{\tilde{\lambda}}}(s_0) \\
&\leq |V_0^{\pi_{\lambda^*}}(s_0) - \hat{V}_0^{\pi_{\lambda^*}}(s_0)| + |\hat{V}_0^{\pi_{\tilde{\lambda}}}(s_0) - V_0^{\pi_{\tilde{\lambda}}}(s_0)| \\
&\quad + |\hat{V}_0^{\pi_{\lambda^*}}(s_0) - \tilde{V}_0^{\pi_{\lambda^*}}(s_0)| + |\tilde{V}_0^{\pi_{\tilde{\lambda}}}(s_0) - \hat{V}_0^{\pi_{\tilde{\lambda}}}(s_0)| \\
&\leq 2 \max_{\lambda \in \Lambda} |V_0^{\pi_\lambda}(s_0) - \hat{V}_0^{\pi_\lambda}(s_0)| + 2 T r_{\max} \sqrt{\frac{\ln(4/\delta)}{2n}} \\
&\leq 2\varepsilon_r T + \varepsilon_p r_{\max} T^2 + r_{\max} T \sqrt{\frac{2 \ln(4/\delta)}{n}}.
\end{aligned}$$

The last inequality follows from Theorem 1. $\qquad \square$

This result is a sanity check that we can reason about error in identifying hyperparameters based on model error. However, it has several limitations. One limitation is that the result is for continuing problems

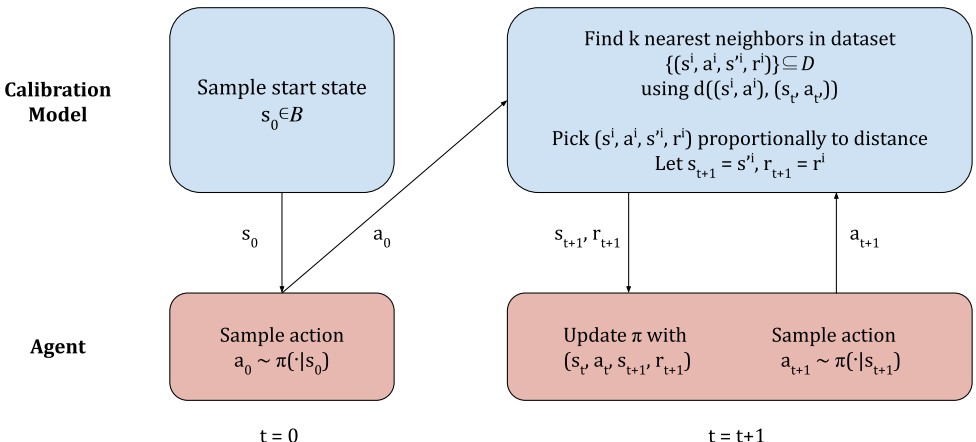

Figure 2: The interaction between the calibration model and the agent in each episode.

and fixed-horizon episodic problems, but not variable length episodic problems. The analysis does not address variable length episodic problems, because it would impact both the histories on which policies are conditioned as well as the definition of the value function for this learning policy. The more important limitation, however, is that the bound depends on the length of interaction $T$, and worse on the squared length of interaction $T^2$ if we assume $v_{\max} \doteq T r_{\max}$. Even if episodes are short, we expect the agent to learn for thousands of steps and so $T$ can be quite large. It may be difficult to obtain sufficiently low $\varepsilon$ (model error) to control for this accumulating error over many steps.

We could potentially obtain a better result by considering smoothness in performance with respect to hyperparameters. Empirical studies suggest performance changes smoothly as a function of hyperparameters, when hyperparameter sensitivity plots are shown. We hypothesize there exists a subset of hyperparameters such that $V_0^{\pi_\lambda}(s_0)$ is smooth w.r.t. the hyperparameters in the subset, and hyperparameters outside the subset have very low $V_0^{\pi_\lambda}(s_0)$. Therefore, the error bound from Theorem 2 just needs to be smaller than the performance gap between hyperparameters in the good subset and hyperparameters outside the good subset to guarantee finding a nearly optimal hyperparameter setting. This direction is an important next step.

## 5 Stable Calibration Models with KNNs

We develop a non-parametric k-nearest neighbor (KNN) calibration model that (a) ensures the agent always produces real states and (b) remains in the space of states observed in the data, and so is stable under many iterations. The idea is simple: the entire offline data log constitutes the model, with trajectories obtained by chaining together transitions. Figure 2 shows the interaction between the calibration model and the agent in each episode. The KNN model is easy to use, which is relevant for those trying to get things working in the real world. In addition, the KNN calibration model is extremely lightweight and allows fast simulation, which is essential for our application: training possibly hundreds of RL agent's with different hyperparameter configurations from scratch.

There are, however, several nuances in using this strategy to obtain calibration models. In particular, the method relies heavily on the distance metric used to find nearest neighbors. Further, the dataset is limited, and may have poor coverage for actions in certain states. We start by introducing the basic approach, and then discuss these two nuances in the following two subsections. We conclude by contrasting the approach to other ways to learn the calibration model, particularly with neural networks.

### 5.1 The KNN Calibration Model

The calibration model needs to produce a (stochastic) outcome next state and reward $r, s'$, given a state and action $s, a$. We can produce novel trajectories from a dataset, by noting that if a state-action pair $(s_t, a_t)$ is

---

**Algorithm 3** Learn KNN Calibration Model

---

**Input:** dataset $\mathcal{D}$ with tuples of the form $(S, A, S', R)$, number of nearest neighbors $k$ (default $k = 3$)
**Constructs:** Representation $\psi$ for distances, KD-Tree $Trees$ for fast nearest neighbors search, $R_{\text{default}}$ default return

   $\psi \leftarrow$ LaplaceRepTraining($\mathcal{D}$) in Algorithm 11
   $Trees \leftarrow$ KDTreeConstruction($\psi, \mathcal{D}$) in Algorithm 5
   Extract starting states $\mathcal{B} \subseteq \mathcal{D}$
   Set $R_{\text{default}}$ to minimum return in dataset (pessimistic default return)

---

**Algorithm 4** Sample KNN Calibration Model

---

**Input:** State $s_t$, Action $a$; if no action is given, procedure returns a start state

   **if** No action is given **then**
      **return** Sample $s \in \mathcal{B}$ uniform randomly
   **end**
   // Find $k$ nearest neighbors to $s_t, a_t$, to get potential next states and rewards
   $(s'_i, r_i, d_i)_{i=1}^k \leftarrow$ KDTreeSearch($\psi(s_t), a, Trees, k$) in Algorithm 7
   // If closest neighbor is far, then return a default return and terminate
   **if** $\min_i d >$ threshold **then**
      $r \leftarrow R_{\text{default}}, s' \leftarrow$ terminal
      **return** $(r, s')$
   **end**
   Sample $i \in [1, k]$ according to probabilities $p_i = 1 - \frac{d_i}{\Sigma_{j \in [1,k]} d_j}$
   **return** $(r_i, s'_i)$

---

similar to $(s_i, a_i)$ for a stored tuple $(s_i, a_i, r_i, s'_i)$, then it is plausible that $r_i, s'_i$ could also have been observed from $(s_t, a_t)$. To allow for stochastic transitions, the $k$ most similar pairs to $(s_t, a_t)$ can be found, and the next state and reward selected amongst these by sampling proportionally to similarity.

More formally, given the current state-action pair $(s_t, a_t)$, the model searches through all tuples $(s, a, s', r)$ and selects the $k$ nearest neighbors, according to similarity between $(s_t, a_t)$ and $(s, a)$. (We discuss how to compute similarity in Section 5.2.) Let $\{(s_i, a_i, r_i, s'_i)\}_{i=1}^k$ correspond to these tuples, and $d_i$ to the distance between $(s_t, a_t)$ and $(s_i, a_i)$. Then these $(r_i, s'_i)$ are all possible outcome rewards and next states, where the likelihood corresponds to similarity to $(s_i, a_i)$. If $(s_t, a_t)$ is very similar to $(s_i, a_i)$, then $(r_i, s'_i)$ is a likely outcome. Otherwise, the more dissimilar, the more unlikely it is that $(r_i, s'_i)$ is a plausible outcome. The tuple $i$ is sampled proportionally to $p_i = 1 - \frac{d_i}{\Sigma_{j \in [1,k]} d_j}$, where a smaller distance indicates higher similarity.

This procedure is summarized in Figure 2. At the start of each episode, a start state $s_0$ is sampled randomly from the set of start states in the dataset. The agent takes its first action $a_0$, to get the first pair $(s_0, a_0)$ and the $k$ nearest neighbors are found. This process continues until the agent reaches a terminal state, or the episode is cutoff and the agent teleported back to a start state. An overview of learning the KNN calibration model is given in Algorithm 3 and sampling the model in Algorithm 4.

There are several details worth mentioning in the algorithms. First, a KNN model relies heavily on an appropriate distance. For example, for input states that correspond to $(x, y)$ position, Euclidean distance can be a poor measure of similarity. If there is a wall in the environment, two states might be similar in Euclidean distance, but far in terms of reachability with notably different dynamics. We ameliorate this by learning a new representation—called a Laplace representation—$\psi(s)$ and using Euclidean distance in this new space that better reflects similarity in transition dynamics, as described in Section 5.2.

Second, there may be no similar pairs in the data for a given $(s, a)$. The state is one that is observed in the dataset, but the action may not be since it is selected by the agent running in the calibration model. When the next outcome state $s_{t+1}$ is chosen from $s_t$, the agent selects action $\tilde{a}_{t+1}$. The dataset might contain multiple transitions from states like $s_{t+1}$—including of course the transition that includes $s_{t+1}$—but

these may be for only a subset of the actions. If none of these transitions uses $\tilde{a}_{t+1}$, then the dataset has insufficient coverage to infer what might occur when taking that action in the environment. When this occurs in Algorithm 4—when the closest point (minimum distance) is too far away (above a threshold)—we set the return to a default return and terminate the episode. We discuss an appropriate choice for this default return in Section 5.3.

Finally, we want to ensure that the model is efficient to query, even if we have a large dataset. For the discrete action setting, it is possible to get an $O(1)$ look-up by caching the nearest neighbors upfront. For $n$ datapoints, for each action we construct a table with $n$ rows and $k$ columns for the nearest neighbors, where each neighbor is stored as its index from 1 to $n$. Each transition consists of jumping between rows, using these indices. The detailed pseudocode is provided in Appendix A.1. More generally, for continuous actions, we can use a k-d tree (Bentley, 1975) to search for the k-nearest neighors. The k-d tree takes the transformed state-action pair, $(\psi(s), a)$ as the key for the search. For a dataset of size $n$, it costs $O(n \log n)$ to construct the k-d tree and $O(\log n)$ to query for a nearest neighbor. This low computational complexity is key to allow us to use all of the data to create our calibration model.

## 5.2 Improving the Distance Metric for the KNN

It is not hard to imagine examples where Euclidean distance on states or observations does not appropriately reflect similarity of states. For example, in a maze environment, if inputs correspond to $(x, y)$, two nearby points in Euclidean distance may actually be on opposite sides of a wall, thus far apart in terms of transition dynamics. Similarly, Euclidean distance does not apply to images, since pixel-wise difference can make every image look very different from all the others in the dataset.

Instead, we exploit a standard approach in metric learning: we first map the inputs to a new space where Euclidean ($\ell_2$) distance is meaningful. In particular, we would like a new representation $\psi(s)$ where states $s_i$ and $s_j$ that have similar outcomes in terms of states and rewards are mapped to similar vectors $\psi(s_i) \approx \psi(s_j)$, and ones with dissimilar outcomes are mapped to different representations.

Such representations that reflect similarity in terms of transition dynamics have been explored under what are called *Laplace representations* (Wu et al., 2019). The approach relies on having a stored trajectory that maintains the order of interaction. The objective includes two components: an *attractive term* that encourages two states that are nearby in the trajectory to have similar representations, and a *repulsive term* that encourages randomly sampled states to have different representations. For a neural network $\psi_\theta$ with parameters $\theta$, the last layer of the NN $\psi_\theta(s)$ has loss

$$\sum_{s_t \sim \mathcal{D}} \|\psi_\theta(s_t) - \psi_\theta(s_{t+1})\|_2^2 + \sum_{s_i, s_j \sim \mathcal{D}} \left( (\psi_\theta(s_i)^T \psi_\theta(s_j))^2 - \|\psi_\theta(s_i)\|_2^2 - \|\psi_\theta(s_j)\|_2^2 \right)$$

The inclusion of representation norms $-\|\psi_\theta(s_i)\|_2^2$ ensures that the representation is not simply decreased to zero to satisfy the first attractive term. Minimizing this objective encourages $\|\psi_\theta(s_t) - \psi_\theta(s_{t+1})\|_2^2$ to be small for states right beside each other in the trajectory—temporally close. The second term $(\psi_\theta(s_i)^T \psi_\theta(s_j))^2$ is the repulsive term that encourage random pairs to have orthogonal representations. It is possible for $s_t, s_{t+1}$ to be randomly selected for the second term, but this is not that likely under the possible $n^2$ pairs; the first term dominates, ensuring these nearby points have similar representations. More details on learning the Laplace representation are given in Appendix A.1.1.

The distance for a state-action pair is defined differently for discrete and continuous actions. For discrete actions, two actions are considered similar only when they are exactly the same. The resulting distance is

$$d((s_i, a_i), (s_j, a_j)) = \begin{cases} d_s(s_i, s_j) & \text{if } a_i = a_j \\ \infty & \text{else} \end{cases} \quad \text{for } d_s(s_i, s_j) \doteq \|\psi(s_i) - \psi(s_j)\|_2^2$$

In practice, we simply keep separate data structures for each action, to find nearest neighbors. For continuous action problems, the Laplace representation can actually be learned on $(s, a)$ directly, to obtain $\psi(s, a)$.

### 5.3 Insufficient Data Coverage

We do not require the dataset to have perfect state and action space coverage. We only query the KNN calibration model from states $s$ that are in the dataset, by construction. But, for a given action $a$, there may be no state-action pair that is similar to $(s, a)$ and so there is insufficient information about the outcome for that pair. What then should the model return?

A natural choice is to truncate the episode, provide a default return—as if the agent had managed to visit future states in the environment—and transition back to the start state. This synthetic interaction in the calibration model is inaccurate, so we encourage the agent to learn within the parts of the calibration model that meaningfully reflect the true environment and avoid these unknown areas. This suggests using a *pessimistic* default return. The default return can be set to the minimal return experienced in the dataset. When the agent reaches these unknown state-action pairs, it receives a low return and on the next episode is less likely to come back to this unknown state-action pair.

Pessimism has also been used in offline RL, but for a subtly different purpose than here. The goal of pessimism in offline RL is to avoid unknown areas, as it is unlikely for the fixed policy to be good in a region that it has never observed and further that unknown region may be dangerous. It is much safer to stay within the data distribution, and simply improve performance within that region.

For us, the policy can adapt online if it reaches unknown areas, so it is not necessary to ensure the agent avoids them in the environment. But, we avoid encouraging the agent to visit these unknown areas in the calibration model because they are not reflective of the true environment, potentially skewing hyperparameter selection. For example, if the agent was instead encouraged to visit these state-action pairs (using optimism), then it might find an unknown state-action pair and spend most of its time visiting it. The hyperparameters would be tuned to work well for this interaction, with short episodes and (default) Monte-carlo return from this state-action that do not require any bootstrapping. Our primary purpose with this choice, therefore, is to make interaction in the calibration model more similar to interaction in the environment, under the unavoidable limitations of insufficient data coverage.

### 5.4 Alternative Calibration Models

We can contrast this KNN calibration model to two natural alternatives: a kernel density estimator (KDE) model and a neural network model. A KDE model is a non-parametric estimator, that has even been investigated for model-based RL (Pan et al., 2018). Like our KNN calibration model, it should also stably remain within the region defined by the dataset. However, unlike the KNN calibration model, a KDE calibration model could produce non-existent states. It effectively interpolates between observed datapoints, and so results in significant generalization. If we consider again the example with $(x, y)$ position in a gridworld with walls, then the KDE calibration model could produce transitions within the wall.

Another alternative is to use a neural network (NN) to learn a calibration model. The dataset can be used to learn the expected next state and reward, for given state and action, using regression on inputs $(s, a)$ and targets $(r, s')$. Or, to obtain a distribution over the next states and rewards, a conditional distribution can be learned using mixture density networks or stochastic networks. Simulators have been learned in RL on real data, particularly with recurrent NNs (RNNs) to handle partial observability, such as for gas turbine control (Schaefer et al., 2007; Schäfer, 2008), drawing on the larger literature using RNNs for system identification (Barabanov & Prokhorov, 2002; Yu, 2004; Mohajerin, 2012).

At the same time, learning transition dynamics with NNs in RL can be challenging, and can cause issues when used as simulators. Such NN models can produce non-existent states, just like the KDE model. With an extremely long rollout, the prediction error accumulates, and the model may generate states that are out of distribution or become unstable. Several works in model-based RL have illustrated that iterating such models can produce less and less plausible outcomes states (Talvitie, 2017; Jafferjee et al., 2020; Abbas et al., 2020; Chelu et al., 2020). This is particularly problematic in the Data2Online setting, where the number of steps of iteration is much larger than what is typically used in model-based RL. The calibration model must mimic the real deployment environment. Initially the agent relies on random exploration to discover good

rewards before learning can adapt the policy, and thus the first few episodes simulated by the model may be extremely long, even thousands of steps.

Avoiding these issues with iterating NN models is an active area of research. One direction for model-based RL has been to train models to be correct over multiple steps of iteration (Talvitie, 2014; Venkatraman et al., 2015; Talvitie, 2017; Williams et al., 2017; Ke et al., 2019). Other work has looked at constraining the architecture to ensure stability (Manek & Kolter, 2019; Lawrence et al., 2020; Takeishi & Kawahara, 2021; Drgona et al., 2022). Such advances are likely to accelerate with the growth in sequence modeling and generation.

In this work, we leverage the power of neural networks to improve the distance metric within our KNN model. Arguably, this distance metric does much of the heavy lifting, with mostly simplistic rules layered on top to transition between samples. The combination allows us to leverage the ability of NNs to scale to high-dimensional inputs, and the simplicity and interpretability of KNNs. It provides an easy-to-use alternative to learning NNs or RNNs from scratch, which can often require significant expertise. And, by design, it is guaranteed to remain stable and only produce states that have been observed. In some cases an end-to-end RNN model may be more effective, nonetheless, this KNN approach with a NN metric expands the types of models users can consider for their application.

## 6 Experiments

We conducted a battery of experiments to provide a rounded assessment of when an approach can or cannot be expected to reliably select good hyperparameters for online learning. We investigate varying the data collection policy and size of the data logs to mimic a variety of deployment scenarios ranging from a near-optimal operator to random data. We explore selecting hyperparameters of different types for several different agents, and investigate a non-stationary setting where the environment changes from data collection to deployment. We begin with the simplest first question: how does our approach compare to simple baselines and with different choices of calibration model type.

To extensively test the reliability of our approach, we deploy the algorithm on variants of Acrobot, Puddle World, and Cartpole (Sutton & Barto, 2018). All three environments are episodic and have a low-dimensional continuous state and discrete actions. Small environments allow extensive experimentation; critical for hyperparameter analysis and achieving statistically significant results. In addition, recent studies have shown that conclusions from small classic control environments match those generated in larger scale environments like Atari (Ceron & Castro, 2021). Experiments were conducted on a cluster and a powerful workstation using $\sim 8327$ CPU hours and no GPUs. Full lists of all the hyperparameters can be found in the appendix.

### 6.1 Experiment 1: Comparing Calibration Models

In this experiment we investigate the benefits of our approach with different choices of model in two classic control environments. We compare our KNN calibration model with learned Laplace similarity metric to an NN model trained to predict the next state and reward given input state and action observed in the calibration data. In addition, we also test an NN calibration model that takes the *Laplacian encoding* (see Section 5) of the current state as input and predicts the next state and reward to provide the network with a better transition-aware input representation. We used two continuous state, discrete action, episodic deployment environments, Acrobot and Puddle World, as described in the appendix and in introductory texts (Sutton & Barto, 2018).

In this first experiment we select the hyperparameters for a linear softmax-policy Expected Sarsa agent (from here on, linear Sarsa) from data generated by a simple policy with good coverage. The agent uses tile coding to map the continuous state variables to binary feature vectors (see Sutton & Barto (Sutton & Barto, 2018) for a detailed discussion of tile coding). This on-policy, Sarsa agent learns quickly but is sensitive to several important hyperparameters. We investigate several dimensions of hyperparameters including the step-size and momentum parameters of the Adam optimizer, the temperature parameter of the policy, and the value function weight initialization. We choose these hyperparameters because their impact on performance is

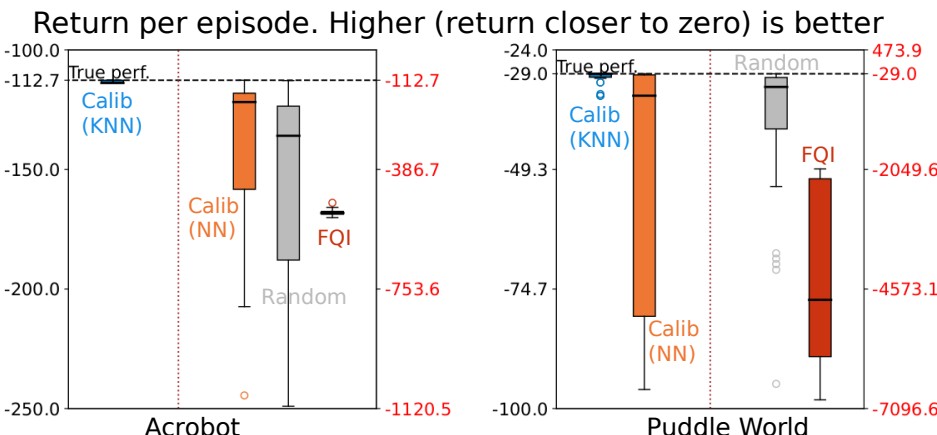

Figure 3: **Hyperparameter transfer with different calibration models.** Each subplot shows the performance of two calibration models compared against two baselines: FQI and random (described in text). The dotted horizontal line indicates the average performance of the best hyperparameter setting in the sweep in the deployment environment. Each box shows the distribution summarizing the true performance in deployment of the best hyperparameters selected in each run of the experiment. We plot the return per episode, thus **higher the better**. The LHS of each subplot uses the LHS y-axis and the RHS (separated by the dotted vertical orange line) uses the RHS red y-axis. In each subplot the bold line represents the median, the boxes represent the 25th and 75th percentiles, and outliers are represented as circles. Low variance indicates that the system reliably chooses the same hyperparameters each run. The performance of the random baseline characterizes the maximum possible variation. Recall the performance of each hyper is precomputed off-line for each hyperparameter combination and is thus not a source of variation in our setup.

somewhat transient and can be overcome by continued learning; this reflects our desire for the agent to continually learn and adapt in deployment.

We used a near-optimal policy for each environment to collect data for building the calibration models. The near-optimal data collection policy for Acrobot can solve the task in 100 steps, and the near-optimal data collection policy in Puddle World achieves an average return of -25. In both cases the policy will provide the system with many examples of successful trajectories to the goal states in the 5000 transition data log.

Our evaluation pipeline involves several steps. First we evaluate the *true performance* (steps per episode for Acrobot and return per episode in Puddle World) of each hyperparameter combination in the deployment environment: running for 15,000 steps in Acrobot and 30,000 steps in Puddle World, averaging over 30 runs. We then use the data collection policy to generate the calibration data log and learn each model. We record the *true performance* of the selected hyperparameters to summarize the performance. This whole process—running the data collection policy to generate a data log, learning the calibration model, and evaluating the hyperparameters—is repeated 30 times (giving 30 datasets with 30 corresponding hyperparameter selections). The statistic of interest is the median and distribution of the *true performance* for the hyper-parameters selected across runs. In the ideal case, if there is one truly best set of hyperparameters, the system will choose those every time and the variance in *true performance* would be zero.

We also included two baselines. The first is randomly selecting hyperparameters, called Random, to get a sense of the spread in performance; all methods should outperform Random. We also include an Offline RL algorithm, called Fitted-Q Iteration (FQI) (Ernst et al., 2005), that learns a policy from the calibration data and then deploys the learned policy fixed in the deployment environment. For the FQI baseline we simply plot the distribution of performance of each of the 30 extracted policies on the deployment environment. For each policy, we average the performance over 30 random seeds. We tested FQI with a tile coded representation and a NN representation; the tile coded version performed best and we report that result.

Figure 3 summarizes the results. In both environments the KNN calibration model performed well, selecting the same hyperparameters as would a sweep directly in the deployment environment. The NN calibration models perform poorly overall. The NN calibration model using raw inputs (no Laplacian encoding) was not as effective, and so we only include results for the NN with the Laplacian encoding in Figure 3 and relegate the other to Appendix D.1. Their performance can be unstable, choosing hyperparameters with

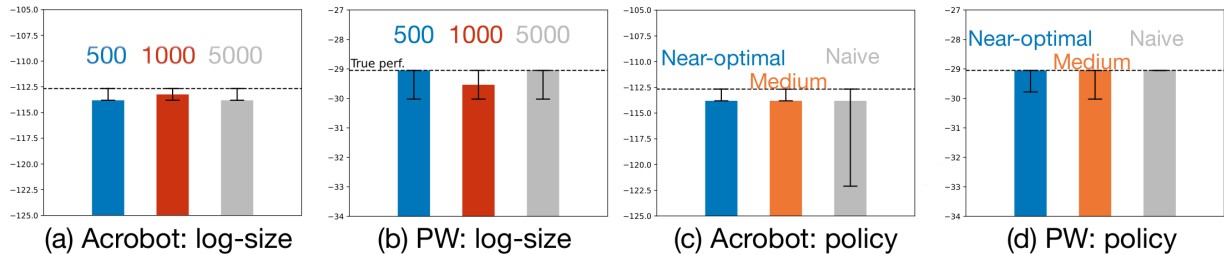

Figure 4: **The role of data logs.** Plots (a) and (b) show the median deployment performance of hyperparmaters selected from calibration models constructed from different sized data logs (with 25% and 75% quartiles). Plots (c) and (d) show the median deployment performance under different policies used to collect data logs for constructing the calibration model. In both subplots, **higher is better**. In these bar plots the median is shown by the top of the colored bar, and the quartiles are shown by the black whiskers. Overall, the utility of the calibration model appears largely insensitive to the data log size and policy in these environemnts; hyperparameters that perform well in deployment can be selected.

good performance in some runs, but often choosing poor hyperparameters. FQI generally performs worse than even Random. Note that we spent quite a bit of time improving FQI, as well as optimizing over several stepsize and regularization hyperparameters. This suggests the calibration data log is too limited to extract a good policy and deploy it without additional learning, but the data appears useful for selecting hyperparameters with the KNN calibration model.

We also used our approach to tune both step-size parameters of a linear Actor-critic agent with tile coding. The KNN calibration model was able to select top performing hyperparameters for Actor-critic in both Acrobot and Puddle World—though the agent performed worse than linear Sarsa (results in Appendix D.2).

## 6.2 Experiment 2: Varying Data Collection Policies

The objective of this experiment was to evaluate the robustness of our approach to changing both the amount of offline data available and the quality of the policy used to collect the data. We experimented with three different policies corresponding to *near-optimal*, *medium*, and *naive* performance to collect 5000 transitions for training our KNN Laplacian calibration model. The near-optimal policy was identical to the one used in the previous experiment. The medium policy was designed to achieve roughly half the visits to goal states after 5000 steps (approximately 90 for Puddle World & 25 for Acrobot) compared to the near optimal policy. The naive policy was designed such that it achieved significantly fewer visits (approximately 35 for Puddle World & 12 for Acrobot). We also tried different data log sizes of 500, 1000, and 5000 samples using the medium policy, all shown in Figure 4.

The results in Figure 4 show that our approach is largely insensitive to data log size and policy in these classic environments. Even 500 transitions contains enough coverage of the state-space and successful terminations to produce a useful calibration model. This is in stark contrast to the FQI results in Experiment 1, where a policy trained offline from the same size data log failed to solve either task. Exploration in both these environments is not challenging; therefore, the success of the calibration model is not surprising. This positive outcome, however, reflects that it may be simpler to pick hyperparameters *in some environments*. In Experiment 4, we investigate a failure case in Cartpole.

## 6.3 Experiment 3: When the Environment Changes

Learning online is critical when we expect the environment to change. This can happen due to wear and tear on physical hardware, un-modelled seasonal changes, or the environment may appear non-stationary to the agent because the agent's state representation does not model all aspects of the true state of the MDP. In this latter case it is often best to track rather than converge; to never stop learning (see (Sutton et al., 2007)). In our problem setting, the deployment environment could change significantly between (a) calibration data collection and (b) the deployment phase. Intuitively we would expect batch approaches that simply deploy

a fixed policy learned from data to do poorly in such settings. The following experiment simulates such a scenario with the Acrobot environment.

The experiment involves two variants of the environment. As before, we collected 5000 transitions using the near-optimal policy in Acrobot and then applied our approach to select good hyperparameters for the linear Sarsa agent. Unlike before, we evaluate the hyperparameters selected on a second, *changed* Acrobot environment. In the changed Acrobot environment we doubled the length and mass of the first link length. Our two phase setup changes the dynamics of Acrobot but does not prevent learning reasonably good policies as you will see in the results below. This whole process was repeated 30 times (generating 30 datasets with a corresponding 30 calibration models) to aggregate the results presented in Figure 5.

We included three baselines to help contextualize the results: (1) transferring the policy from the first environment, (2) transferring the policy learned in the calibration model, and (3) FQI. The first baseline, called *Sarsa (True)*, simply transfers the policy learned in the first Acrobot environment to the changed Acrobot environment (no calibration model was used, hence the label *true*). The second baseline, called *Sarsa (Calibration)* simply uses the best performing policy learned by Sarsa in our calibration model, where the calibration model is created with data from the first Acrobot environment. Finally, we also included a FQI baseline. We trained a policy using FQI with tile coding on the data generated from the first environment (the same data used to build the calibration model). Then we evaluated the policy learned by FQI on the changed Acrobot environment. These baselines are meant to illustrate how performance might be effected if the environment dynamics changed but a prelearned policy was applied without taking the changes into account, perhaps because no one noticed the environment had changed. In all three baselines the policy evaluated in the second environment is fixed (no learning in deployment).

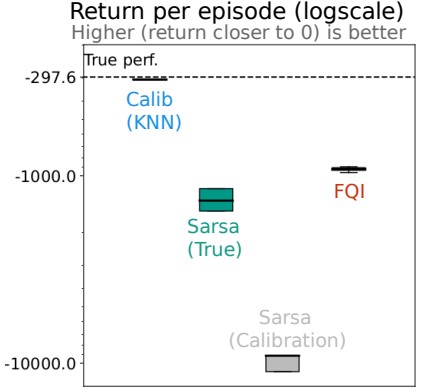

Figure 5: **Selecting hyperparameters in the face of non-stationarity.** The plot above summarizes the performance of our approach compared with three fixed-policy transfer approaches (described in text). Selecting hyperparameters for deployment works well even when the environment changes between calibration data collection and deployment. Deploying fixed policies, on the other hand, performs poorly by comparison.

The results in Figure 5 highlight that transferring fixed policies can be problematic when the environment changes. Our calibration-based approach performs best and appears robust under the abrupt non-stationarity tested here. Clearly, the difference between the two environments is significant enough such that transferring a policy directly learned on the first environment (the Sarsa-True baseline) performs worse than using our approach to select hyperparameters in the calibration model and then learning a new policy from scratch. Interestingly, learning and transferring a policy from the calibration model was worse than using than training on the first environment or training from the calibration data (as in FQI). It is not surprising that transferring hyperparameters and learning in deployment is more robust than transferring fixed policies, in these non-stationary settings.

## 6.4 Experiment 4: A Failure Case

Our approach is not robust to all environments and data collection schemes. In this section we investigate when it can fail. One obvious way our approach can fail is if the agent's performance in the calibration model is always the same: no matter what hyperparameter we try, the system thinks they all perform similarly. To illustrate this phenomenon we use the Cartpole environment. In Cartpole, the agent must balance a pole in an unstable equilibrium as long as it can. If the cart reaches the end of the track or the pole reaches a critical angle, failure is inevitable. Near-optimal policies can balance the pole for hundreds of steps rarely experiencing failures and, thus, visit only a small fraction of the state-action space. A data log collected

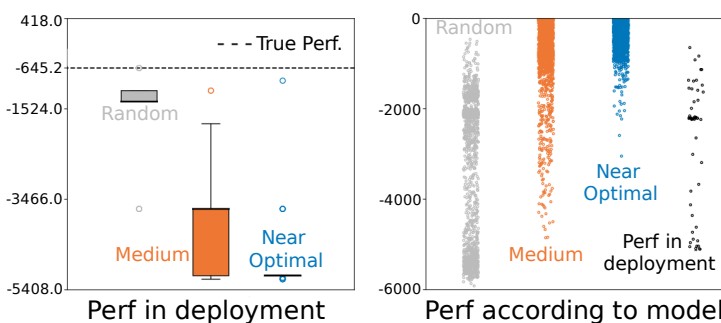

Figure 6: **Success and failure in Cartpole.** This plot shows performance of three different calibration models constructed from random, near-optimal and medium policies. **Left**: the performance of the hyperparameters in deployment as picked by different calibration models. **Right**: each model's evaluation of all hyperparamters across all 30 runs. Ideally the distribution of performance would match that of the hyperparameter performance in the deployment environment—black dots far right.

from the near-optimal policy would likely produce a calibration model where failures are impossible and all hyperparameters appear excellent.

We test this hypothesis by looking at three data collection policies. We used a random policy, a near-optimal policy with random initial pole angles, a medium policy that was half as good as the near-optimal policy (twice as many failures). We expect the random policy to provide useful data for the calibration model, whereas the near-optimal policy will cause failure due to the above reason. The interim policy helps understand sensitivity to this issue: we might expect the issue to be resolved with a less optimal policy.

Figure 6 indeed shows that the dynamics of Cartpole combined with particular data collection policies can render the calibration model ineffective We see in the left-hand plot that the hyperparameter chosen with the calibration model using random data performs somewhat reasonably, but fails for both the medium and near-optimal policies. Even with random starting states the calibration model for near-optimal policy: the calibration model never simulated dropping the pole. The random policy produced the best calibration model. Unsurprisingly, the random policy drops the pole every few steps and thus the log contained many failures and higher state coverage. Nevertheless, the performance was still poor because there were no examples of balancing the pole for many consecutive steps: the model constructed from random data was still a poor model of the true deployment environment.

We can see this further, by looking at the performance estimates under the three calibration models. The right-hand plot in Figure 6 shows the performance of all the hyperparameters according to the calibration model (before deployment). The blue dots show that most hyperparameters appear good in calibration when the model is constructed with data from a near-optimal policy. At the other extreme the grey dots show a large spread in performance of hyperparameters when the model is constructed with data from a the random policy. Note that none of the grey dots appear as low on the y-axis compared with the blue and orange dots corresponding to the other two policies. This indicates that calibration with the medium and near-optimal policy models incorrectly inflate the performance of many hyperparameter combinations, whereas calibration with the random-policy model potentially undervalues some hyperparameter combinations.

One could certainly argue that many applications might not exhibit this behavior—especially since it is largely caused by a task with two modes of operation (failing or balancing). Extracting a policy initialization from the calibration data (perhaps via behavior cloning) and then using this initial policy in both hyperparameter selection and deployment could avoid these problems in Cartpole, but we leave these experiments to future work. Regardless, this experiment provides an important reminder that we will not anticipate all situations in the deployment of RL in the real world; there is no general black-box strategy for deployment and failures will happen.

# 7 Moving Beyond Grid Search

The calibration model is an offline artifact that we can use as we like without impacting the deployment environment. We can use the model in smarter ways to discover and evaluate good hyperparameters for deployment. In fact, we can leverage the large literature on *algorithm configuration*, which explicitly deals with efficient methods to search over hyperparameters. In this section, we explain how to incorporate these approaches and test two strategies, as a replacement for grid search.

## 7.1 Improving the Hyperparameter Search

A variety of hyperparameter search approaches were introduced under sequential model-based optimization (SMBO) (Hutter et al., 2011), but methods built on Bayesian optimization (BO) (Snoek et al., 2012) have become more popular. Complementary to these approaches are those that direct computation to promising hyperparameters and stop performance evaluations of poor hyperparameters early, as in Hyperband (Li et al., 2018), or that design the algorithm to do both (Klein et al., 2017; Falkner et al., 2018). All these approaches attempt to find the maximum of the performance function, assuming that function is expensive to query.

BO algorithms approximate the performance function $f(\lambda)$, and use this approximation to decide what hyperparameter setting $\lambda$ to test next. The general approach is to (1) maintain a posterior distribution over the performance function $f$, (2) find a candidate set of optimal hyperparameters $\lambda_c$ according to a criteria like expected improvement under the current posterior over $f$, (3) evaluate $\lambda_c$, obtaining $y = f(\lambda_c)$ and (4) update the posterior with sample $(\lambda_c, y)$. Once the algorithm terminates—typically by reaching a time limit—the best $\lambda_c$ out of all the candidates tested is returned, according to the maximal $y$. The primary purpose of learning the posterior $f$ is to direct which $\lambda_c$ should be tested, though some algorithms do solve an optimization at the very end of this procedure to find $\lambda$ with the maximal posterior mean (see (Frazier, 2018) for a nice overview).

Due to the importance of hyperparameter optimization for machine learning—in a growing field called Auto ML—the development of BO methods has been focused on large numbers of hyperparameters, for training large models on large datasets. For this highly expensive setting, it is worth carefully crafting advanced algorithms that minimize the need to train and evaluate large models. These complex methods can then be released within packages, to facilitate their use, as they may be difficult to implement from scratch.

**BO in our experiments:** We use an open-source package (Nogueira, 2014–), which uses gaussian processes for optimizing the hyperparameter setting. We chose to use upper confidence bounds, with a confidence level of 2.576—the default in the package—as the acquisition method. The queue is initialized with 5 random samples and the algorithm is run for 200 iterations.

For our setting, each evaluation is not as complex and we need not use such advanced approaches. Instead, our primary goal is simply to answer: if we allow hyperparameters to be optimized over a continuous set, can we improve on a basic grid search? For this question, we also test two simple approaches: random search and the cross-entropy method (CEM). Random search involves simply testing a fixed number of hyperparameter settings, and selecting the one with maximal performance. Though simple, it is a common baseline because it has been shown to be competitive with more complex search strategies (Bergstra & Bengio, 2012).

CEM (Rubinstein, 1999) is an approach to global optimization of functions, like BO, but is based on a simpler strategy of (1) maintaining a distribution over the inputs (hyperparameters), (2) increasing the likelihood of the top percentile of sampled values under this distribution, according to the performance function. The distribution is simple to sample, and the percentile easy-to-compute, making this approach simpler to use than BO.

CEM has not been used for hyperparameter optimization, to the best of our knowledge. Likely the reason is that BO strategies provide a more powerful way to carefully reason about what candidate points to sample. CEM instead slowly narrows a distribution over hyperparameters, and does not reason about confidence intervals nor about a criterion (acquisition function) to identify ideal candidates. Nonetheless, we include it as a simpler strategy, to investigate performance of using calibration models with a hyperparameter search

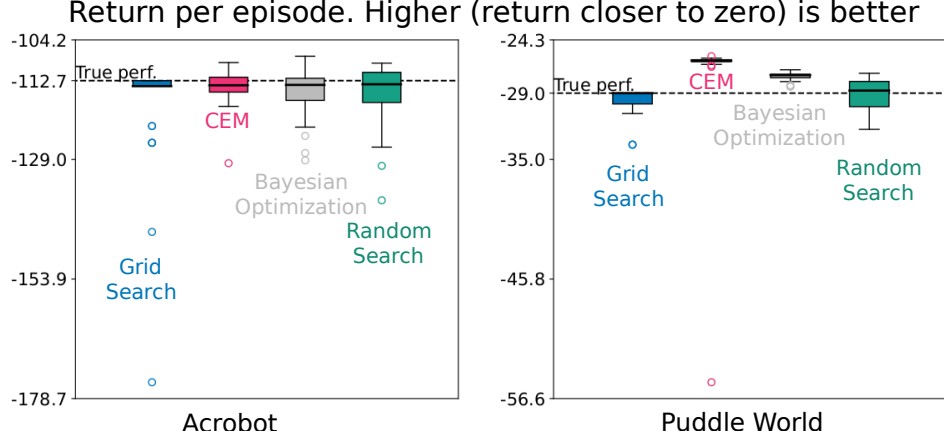

Figure 7: **Finding good hyperparameters via black-box optimization in calibration.** Here we compare four different strategies for optimizing hyperparameters: (1) grid search (which we have used in all previous experiments), (2) random search, (3) our CEM procedure, and (4) the Bayesian optimization (BO) approach. The y-axis is same as the one in Figure 3. Generally, all four approaches perform well highlighting the robustness of using the calibration model and transferring hyperparameters to deployment. Random search is better or comparable to grid search, but this is not a surprising as random search is typically found to be a strong baseline in hyperparameter optimization. Note the *True performance* in the plots above represent the deployment performance of the best hyperparameter combination from a discrete set; the same set used by grid search. CEM, BO and random search can obtain higher deployment performance because they search a continuous range and thus find better hyperparameter settings. This is one of the major benefits of using better optimization methods in calibration than grid search.

in-between naive random search and the more advanced BO search. We emphasize that it is not critical which hyperparameter optimization approach is used within our framework; any method can be swapped in.

**CEM in our experiments:** Our setting has two nuances compared to the typical setting where CEM is used: our function is expensive to query and we only get a stochastic sample. We provide a modified CEM algorithm, that still reflects the general framework for CEM, but using an incremental update—similar to stochastic gradient descent—to account for the stochasticity in our function query. The algorithm is summarized in Algorithm 13 in Appendix C.1.

## 7.2 Experiment 5: Hyperparameter Tuning with Alternative Optimization Approaches

In this section we compare grid search, random search, Bayesian optimization, and our simple CEM approach for hyperparameter selection. Are these approaches interchangeable in our setup? Does searching a continuous hyperparameter space result in performance gains and at what cost? The goal of the experiment is to highlight that alternative hyperparameter optimization approaches beyond a basic grid search are possible and to investigate if they are beneficial.

In this experiment, we use the same settings as above, but now optimize the temperature $\tau$ and stepsize $\alpha$ as continuous values in the ranges [0.0001, 5.0] and (0.0, 0.1] respectively for Acrobot, and [0.0001, 10.0] and [0.0, 1.0] respectively for Puddle World. The random search approach simply generates $k$ possible hyperparameters from the continuous ranges above and evaluates each in parallel in the calibration model. The best performing hyperparameters according to the calibration phase are used in deployment. Both random search and CEM use 100 iterations, to make computation used comparable to grid search, while Bayesian optimization uses 200 iterations.

Random search, Bayesian optimization and CEM outperform grid search, as we can see in Figure 7. The performance improvements are especially stark in Puddle World. Even when tuning only on the calibration model, the agent can outperform the best hyperparameters found by a grid search on the true environ-

ment. This is why the return for CEM is higher than the dotted line showing the performance of the best hyperparameters within the set used for the grid search. These results are promising, in that they show more carefully optimizing hyperparameters on the calibration model helps rather than hurts. A possible hypothesis, apriori, could have been that optimizing more carefully to the calibration model could cause odd or very poor hyperparameters to be chosen and that the restricted set in the grid search actually helped avoid this failure. These results suggest otherwise, and in fact highlight that our previous results could have been produced even more consistent performance with a smarter hyperparameter algorithm.

This experiments in this paper highlight the generality and flexibility of our approach. The calibration model can take different forms. Data collection can be done in a number of different ways. Hyperparameters can be systematically searched or optimized. In the end, numerous other specializations and algorithmic innovations could be included to further optimize performance in real deployment scenarios.

## 8 Conclusion

In this work, we introduced the Data2Online problem: selecting hyperparameters from log of data, for deployment in a real environment. The basic idea is to learn a calibration model from the data log, and then allow the agent to interact in the calibration model to identify good hyperparameters. Essentially, the calibration model is treated just like the real environment. We provide a simple approach, using k-nearest neighbors, to obtain a calibration model that is stable under many iterations and only produces real states. We then conduct a battery of tests, under different data regimes.

Naturally, as the first work explicitly tackling this problem, we have only scratched the surface of options. There is much more to understand about when this strategy will be effective, and when it might fail. As we highlight throughout, this problem should be more feasible than offline RL, which requires the entire policy to be identified from a log rather than just suitable hyperparameters for learning. Our own experiments highlight that offline methods that attempted to learn and deploy a fixed policy performed poorly, whereas identifying reasonable hyperparameters was a much easier problem with consistently good performance across many policies and even small datasets. Nonetheless, we did identify one failure case, where the data resulted in a model that made the environment appear too easy and so most hyperparameters looked similar. Much more work can be done, theoretically and empirically, to understand the Data2Online problem.

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
