# OpenReview forum: "No More Pesky Hyperparameters: Offline Hyperparameter Tuning for RL"
_TMLR — Accepted by TMLR_

### Review · Reviewer_MhrF · 2022-06-02

**Summary Of Contributions:**

The paper deals with the possibility to choose hyperparameters for reinforcement learning based on a model of the environment. This model is created from pre-recorded data of the environment. This model is called "calibration model" in this paper, which I think is a very appropriate name.

One problem is that the paper presents this technique as being new, although it is state of the art.
Especially the insight "The calibration model need not be a perfect simulator to be useful for identifying reasonable hyperparameters, whereas learning a transferable policy typically requires accurate models.", though completely correct and very important in practice, is not a new insight.

However, since there is little published systematic research on this technique, this paper can help to fill this gap.

In this paper, experiments are performed on two very simple benchmarks.
As a data-based model, a k-nearest neighbors (KNN) approach improved by metric learning is used and compared to a NN that also uses the variables of the learned metric.
Furthermore, some limitations are discussed and theoretical considerations are provided.


**Broader Impact Concerns:**

No concerns.

**Requested Changes:**

The paper can contribute to scientific progress 1) by naming and discussing the known approach for which published studies are scarce, and 2) by presenting KNN with with Laplace representation. Where I consider 1) to be the more important contribution.

However, to be valuable, all inaccurate or insufficiently substantiated claims (see weaknesses) must be reformulated so that they are true and sufficiently substantiated. In particular, because this means not making the claim that the approach is generally novel, it is necessary to completely reword the Abstract, Introduction, and other parts of the text. Possibly, the title should also be chosen more modest and appropriate to the contribution actually made.

Extensive experiments in more difficult and high dimensional environments and comparisons with state of the art system identifiers would drastically increase the usefulness of the paper. However, I think it is understandable if this is seen as future work.


**Strengths And Weaknesses:**

**Strengths**\
The topic is important.\
There is a lack of published studies on experience with this technique.\
The use of KNN models improved by metric learning using Laplace representation as calibration models is, as far as I know, new.\
The term calibration model, which as far as I know is introduced in this paper is very fitting.

**Weaknesses**\
The biggest problem in my opinion is that the paper presents the discussed technique as generally new although it is state of the art.
The selection of appropriate reinforcement learning algorithms (and this includes the hyperparameters of the algorithms) based on a model learned on real data is known from patent specification US8099181 [1], for example.
Furthermore, I know this approach as common practice in the application of RL to real applications.

[1] https://patentimages.storage.googleapis.com/c5/5d/d6/b03f78fa17ce13/US8099181.pdf

Several statements are made for which no sufficient evidence is given in their generality:

•	The statement „We propose a new approach to tune hyperparameters from offline logs of data, to fully specify the hyperparameters for an RL agent that learns online in the real world“ is correct, if one thinks of the specific design of the KNN with Laplace representation. However, since the statement continues with the sentence „The approach is conceptually simple: we first learn a model of the environment from the offline data, which we call a calibration model, and then simulate learning in the calibration model to identify promising hyperparameters“, it is claimed that the whole approach of selecting hyperparameters for reinforcement learning based on a model of the environment, where this model is created using pre-recorded data of the environment, is novel, which is not the case.

•	This is repeated with „We propose a novel strategy to use offline data for selecting hyperparameters“. Again this sentence is correct, if one thinks of the specific design of the KNN with Laplace representation. By the subsequent sentence  „The idea is simple: we use the data to learn a calibration model, and evaluate hyperparameters in the calibration model“ it is again claimed that the approach is new in general.

•	Likewise „Though this is a simple and natural idea, to the best of our knowledge, it is the first general approach for Data2Online“. This approach already existed before. New is the special design of the calibration model as KNN with Laplace representation. Then, in the subsequent sentence, it is also admitted that the use of models is known: „It is common in reinforcement learning to learn models for offline policy evaluation or for planning“. The justification why the own approach is new „These approaches, however, do not need to tackle a key problem we consider in this work: iterating the model for thousands of learning steps.“ is not convincing. 1. why should thousands of time steps be necessary in general? After all, the data on which the calibration model was learned is available, so one can always start from a randomly selected real state from that data set during learning and then iterate the model, say, only 100 time steps. 2. it is not shown that the proposed model is better suited for thousands of iterations than established methods of system identification, like RNN, e.g. [2], [3], and references therein. My conviction is that RNN are much better able to approximate the environment than KNN with Laplace representation. The performed comparison with NN is not sufficient. First, the NN used are not state of the art, in particular not RNN, and second, the environments studied are extremely low dimensional, with very simple dynamics. In my experience, a KNN based approach will be structurally inferior to an RNN, especially in high dimensionality.

•	The statement „There is only one other work considering how to use offline data to evaluate an online agent (Mandel et al., 2016)“ is not true in this form. See [4], [5], [6]

•	Also irritating is the sentence „We are hopeful that, with more research, NN models will become a viable choice for learning calibration models“. NN are an established technique of system identification [2], [3] and references therein, which has also been used in high dimensional environments. In the present work, no sufficient evidence was brought that the proposed KNN based models have advantages over state of the art RNN.

One more note to avoid misunderstandings: NN and RNN are not generally limited to identifying deterministic systems: ensembles and NN with stochastic components (e.g., Bayesian Neural Networks [7]) have been used successfully in stochastic environments.

[2] J. Sjöberg, H. Hjalmarsson, L. Ljung, Neural Networks in System Identification, 1994\
[3] D Ha, J Schmidhuber, Recurrent world models facilitate policy evolution, 2018\
[4] T. Gabel, M. Riedmiller, Reducing policy degradation in neurodynamic programming, 2006\
[5] M. Migliavacca et al., Fitted policy search: Direct policy search using a batch reinforcement learning approach, 2010\
[6] A. Hans, S. Duell, S. Udluft, Agent self-assessment: Determining policy quality without execution, 2011\
[7] S Depeweg, Modeling Epistemic and Aleatoric Uncertainty with Bayesian Neural Networks and Latent Variables, 2018

---

> ### Author Response · Authors · 2022-06-05
> **Author response P1**
>
> Thank you for taking the time to read our paper and provide detailed feedback. We will attempt to respond item by item to your concerns.
>
> **Novelty of approach (First three points raised by R1)**
> *Point #1: Citing patents*
>
> As far as we know, it is not common to cite US patents in academic work. The requirements for the two types of documents (papers versus patents) are quite different. We do acknowledge that the idea itself is highly natural. We are happy to add that it is so natural it is likely already being used in practice, and that this work explores this simple idea in-depth. We can engage with the TMLR organizers to see what they suggest as best practice here.
>
> *Point #2:*
> >“it is claimed that the whole approach of selecting hyperparameters for reinforcement learning based on a model of the environment, where this model is created using pre-recorded data of the environment, is novel, which is not the case.”
>
>
> We did not intend to claim this. The main focus of the work is selecting hyper-parameters for learning from scratch (or warm starting) in deployment. We do not intend to suggest that constructing models of the environment from pre-recorded data is new. It's the application of the model (selecting hypers for learning from scratch in deployment) that is of interest to us.
>
> *Point #3*: Restating of point #2 by quoting from the paper
> “The idea is simple: we use the data to learn a calibration model, and evaluate hyperparameters in the calibration model”.
>
> However, the rest of the paragraph goes on to limit the scope:
> 	“Learning online in the calibration model mimics learning in the environment, and so should identify hyperparameters that are effective for online learning performance.’’

---

> > ### Author Response · Authors · 2022-06-05
> > **Author response P2**
> >
> > # Iterating NN models
> >
> > **R1 wonders:**
> > >“why should thousands of time steps be necessary in general? After all, the data on which the calibration model was learned is available, so one can always start from a randomly selected real state from that data set during learning and then iterate the model, say, only 100 time steps.”
> >
> > Recall that we use our calibration model to mimic the real deployment environment. That means we start an RL agent from scratch training in the calibration model as if it were the environment. Such agent’s often rely on random exploration to discover good rewards before learning can adapt the policy. Therefore, the first few episodes might be extremely long before a goal state is found (e.g., in simple mountain car it takes over 65,000 steps to wander to the goal with a good linear function approximation agent. DQN is much less data efficient). The reviewer proposes simply cutting off episodes (as common in deep RL). However, this is not a universal solution because some deployment scenarios may not allow short cutoffs. In continuing problems there are no terminations at all. Finally, cutting off episodes typically makes the exploration problem much easier and thus may indeed impact the hyperparameters selected. We want our calibration model to reflect the real-world as best we can, and thus we take the design aesthetic to not rely on cutoffs which have many consequences and are not always feasible (e.g., a fusion reaction or stabilizing the Laser Interferometer Gravitational-Wave Observatory (LIGO) during large-scale physics experiments.
> >
> >
> > R1 Suggests:
> >  	>“it is not shown that the proposed model is better suited for thousands of iterations than established methods of system identification, like RNN, e.g. [2], [3]”
> >
> > We think there is a misunderstanding here. We do not intend to denigrate the ability of NNs to model complex systems. We were simply reflecting the widely held notion in RL that iterating NN models is known to be challenging in model-based planning with learned models. We in fact started with NNs and ran into these same issues when modeling our water treatment plant data. The model under iteration would produce invalid states causing issues with the learning agent and thus hyperparameter selection. We could not include such results because of our industrial partner. Nevertheless, we found the same to be true in simulation tasks which inspired our KNN calibration model. **We will sharpen this statement in the text to both report our specific challenges with NN-based calibration models and link it to the wider literature in model-based RL.**
> >
> > We would like to add that we do in fact compare to using NN calibration models and recognize the utility of NNs for improving the calibration model. NNs themselves factor heavily into our KNN solution, because we use NNs to learn the similarity metric. We exploit the utility of NNs to learn on higher-dimensional data–for scaling–and KNN for stability. In our experiments, we found our KNN approach to be much more effective than the NN. This may not always be true, but we did attempt to provide some evidence for our claim that a KNN calibration model could be beneficial.

---

> > > ### Author Response · Authors · 2022-06-05
> > > **Author response P3**
> > >
> > > R1 believes:
> > > >“The performed comparison with NN is not sufficient. First, the NN used are not state of the art”.
> > >
> > > We do not agree. In fact, our network is as SOTA as recent ones used by Deepmind to learn to control fusion reactions. We used Xavier initialization, relu activations, the Adam optimizer and tested a variety of layer sizes and depths. We had also tested improvements, such as those for model iteration, and other architectures, such as with bottleneck layers. In general, the space of NN architectures is vast, and we do not claim any NN could not have out-performed our KNN calibration model. However, the KNN does provide another option for those considering a calibration model approach, and one that might be easier to use. We expand on this further below.
> > >
> > > **We will add a detailed discussion in the paper about our efforts to construct NN calibration models.** Nonetheless, as you highlighted in your review by quoting our paper
> > > “We are hopeful that, with more research, NN models will become a viable choice for learning calibration models”.
> > > We believe the main insight of this work is using models of the environment to select hyperparameters of learning in deployment. We used a KNN model due to our struggles with NN models on our real plant data. We thought this was useful to report to the community; many struggle to train NN models (perhaps lacking industry scale compute or extensive expertise in NN training)---**the KNN model was easy to use and worked well which we think is relevant for those trying to get things working in the real world. In addition, the KNN calibration model is extremely lightweight and allows extremely fast simulation, which is essential for our application: training possibly hundreds of RL agent’s with different hyperparameter configurations from scratch.**
> > >
> > > We would be happy to investigate the SOTA architectures the reviewer suggests, as we are more than happy to use NN calibration models where effective. Supplied reference [3] uses a VAE which we agree are capable of impressive feats. Such approaches are extremely promising, but are largely used in image based domains (where NNs are extremely effective due to convolutions and decades of developments from computer vision research). Nevertheless, we are happy to try this, or another RNN architecture.
> > >
> > > We would like to note that claims of [3] are strongly questioned by follow up work [van Hasselt et al, 2019]. They demonstrated vastly superior performance compared with [3] by simply increasing the number of replay steps in a Rainbow agent, showing that just using the data was better than whatever benefit’s [3]’s model was conveying.
> > >
> > > [van Hasselt et al, 2019] When to use parametric models in reinforcement learning? Hado van Hasselt, Matteo Hessel, John Aslanides

---

> > > > ### Author Response · Authors · 2022-06-05
> > > > **Author response P4**
> > > >
> > > > # On the use of RNNs for calibration
> > > > >“In my experience, a KNN based approach will be structurally inferior to an RNN, especially in high dimensionality.”
> > > >
> > > >
> > > > We agree it is widely held that KNNs struggle with increasing input dimension. This is exactly why we used the Laplace representation–learned with a neural network–to tackle the issues of computing naive Euclidean distances on high dimensional vectors. We combine the scaling benefits of NNs with the simplicity of KNNs. This combination was critical to make the KNN calibration model effective.
> > > >
> > > > As for RNNs versus KNNs, here you might simply have meant neural networks versus nearest-neighbors-based approaches. If so, hopefully the above paragraph clarifies. Otherwise, if you meant RNNs more specifically, with recurrence, then please do say and we can discuss further.
> > > >
> > > > # Offline data for online learning
> > > >
> > > > “The statement „There is only one other work considering how to use offline data to evaluate an online agent (Mandel et al., 2016)“ is not true in this form. See [4], [5], [6]”
> > > >
> > > > We apologize for our potentially imprecise terminology here that may have caused confusion. For us, an online agent is an agent that learns online (in deployment), rather than a fixed policy. (We will more clearly state this). It looks like the references [5] and [6] are focused on evaluating a fixed policy, learned offline; this is the standard goal in off-policy policy evaluation (also called offline policy evaluation). The focus of our paper is using off-line data to improve online learning. For this setting, to the best of our knowledge, only Mandel et al. have considered how to use offline data to answer: how would this algorithm perform if it learned online?
> > > >
> > > > Reference [4] explores an architecture that stores previous data while continually interacting with the world learning in deployment. We agree this approach utilizes data to improve agent performance in deployment. Note there are interesting differences. [4]’s system does not have data collection, calibration and then deployment phases they are all one. In fact, [4]’s approach and more generally Riedmiller’s fitted architectures over the years directly inspired DQN’s use of experience replay and target networks. In this way [4] is as related as DQN to our work. Both attempt to learn a near optimal policy while interacting with the world, by leveraging previously observed data. Our setting is subtly and we argue interestingly different:
> > > > 1. We are provided logs of operator data
> > > > 2. We must design an agent from the data
> > > > 3. That agent must effectively and efficiently learn in deployment on the real plant
> > > >
> > > > Our focus is on a substep of #2: how to select the hyperparameters to improve step #3. In step #2 we could use [4]’s agent architecture, and of course in step #3. We could use DQN. Regardless the point is the same DQN or [4] have important hyperparameters that must be set so they will perform well in step #3.

---

> > > > > ### Comment · Reviewer_MhrF · 2022-07-07
> > > > > **Further remarks**
> > > > >
> > > > > Even though I actually want to praise the outstandingly good rebuttal, I still want to warn against loose generalizations, they can be very harmful. For example, [van Hasselt et al, 2019] does not show "vastly superior performance" in general, but only on Atari2600 benchmarks.
> > > > > As for the problem of invalid states after long rollouts with NN, I can report from experience that this does not happen so easily especially when using RNN (unfolded in time), since RNN are able to learn a real system identification. Furthermore, the production of invalid states can often be avoided by coding corresponding constraints directly in the neural architecture.

---

> > ### Comment · Reviewer_MhrF · 2022-07-07
> > **on P1-P4**
> >
> > First of all, I would like to thank the authors for their extraordinarily detailed answers.
> > If these differentiated explanations (as far as possible) find their way into the text, I think the paper will be very valuable.
> > As before, I disagree on a few points, but that is not the point. What is important to me is that all claims are supported by evidence, that the existence of different approaches and previous work is named and that a discussion takes place, as the authors have shown in the answers.

---

> > > ### Comment · Reviewer_MhrF · 2022-07-08
> > > **Some remarks on RNN**
> > >
> > > In the 90s, RNNs were analyzed a lot and also studied for system identification. A central work is [1].\
> > > A nice overview of the historical development and literature is given in [2].\
> > > Independently of the partially observable aspect, RNN have been used to reliably represent the dynamical behavior, including stability considerations, e.g. [3], or for the identification of chaotic systems [4].\
> > > Control has also been addressed in [1], also in [5].\
> > > Early works on RL were [6] and [7], where the partially observable aspect, the system identification aspect and the use as a model for model based RL are included.\
> > > In [8], very similar to the present work, a data-based model is learned using real data, and further used as an iterated simulation model to compare different RL algorithms. As I understand it, this simulation model (I guess you could call it a calibration model) is iterated over thousands of time steps.
> > > As can be read in one of the co-authors work [9], this calibration model is also an RNN.
> > >
> > > As I notice now, this shows another advantage of the kNN which is used as calibration model in the present work:
> > > In [8], both the calibration model is an RNN, as well as the RL algorithms under investigation.
> > > It is expected that when the calibration model is an RNN, RNN based RL algorithms perform better than other approaches in a kind of overfitting. If one uses a kNN as the calibration model, then it will probably be structurally sufficiently different from the neuro-based RL algorithms used, so that there is less risk of overoptimistic results due to overfitting effects.
> > >
> > > [1] P.J. Werbos, Backpropagation through time: what it does and how to do it, 1990\
> > > [2] N. Mohajerin, Identification and Predictive Control Using Recurrent Neural Networks, 2012\
> > > [3] N.E. Barabanov and D.V. Prokhorov, Stability analysis of discrete-time recurrent neural networks., 2002\
> > > [4] W. Yu, Nonlinear system identification using discrete-time recurrent neural networks with stable learning algorithms, 2004\
> > > [5] A. Delgado, et al., Dynamic recurrent neural network for system identification and control, 1995\
> > > [6] P.B. Backer, The State of Mind Reinforcement Learning with Recurrent Neural Networks, 2004\
> > > [7] A.M. Schaefer, Reinforcement Learning with Recurrent Neural Networks, 2008\
> > > [8] A.M. Schaefer et al., A neural reinforcement learning approach to gas turbine control, 2007\
> > > [9] D. Schneegass, Steigerung der Informationseffizienz im Reinforcement-Learning, 2008, p. 101

---

### Review · Reviewer_3uTX · 2022-06-05

**Summary Of Contributions:**

This paper presents an offline hyperparameter selection scheme based on a non-parametric calibration model for offline RL with online adaptation. The authors present a calibration model trained with KNN, simulate learned policies under the learned calibration model and then pick the hyperparameter that yields the best simulated performance. The authors compare the method to a calibration model trained with neural networks, prior offline RL method with offline tuning scheme (FQI) and offline RL with random hyperparameter selections and show that the proposed method achieves better performance in two control tasks with varying data quality.

**Requested Changes:**

See the above section.

**Strengths And Weaknesses:**

I think the idea of the method is intuitive and easy to follow. It is very natural to consider a model-based offline hyperparameter selection rule. The method is well-motivated and the empirical evidence shows that the proposed method is sensible and can outperform random hyperparameters, offline RL with offline hyperparameter tuning based on Q-values and offline RL with NN-parameterized dynamics models.

However, I do have several concerns, which are listed as follows. First, I think it is a bit unclear why the authors consider the offline-to-online setting as the whole experiment section. The method seems to focus on hyperparameter selection for offline RL without using any online interactions and I don't see why this method is particularly fitted to the offline-to-online scenario. It may be able to apply to the setting where we train RL policies without different hyperparameters either online or offline and then perform offline hyperparameter selections, but I'm not sure if the case where we train the policy online and then select hyperparameters offline is necessary.

Moreover, I think the experiments need to include more comparisons such as [1] and challenging domains such as DM control, DM locomotion, Manipulation playground and tasks from the D4RL benchmark, which will be pivotal to see the effectiveness of the approach.

Finally, I believe that it is important to test the effectiveness of the method on different (offline) RL algorithms. It would be particularly interesting to test how the calibration models help pick hyperparameters for standard offline model-free RL algorithms (e.g. CQL, IQL, etc.) and model-based methods (e.g. MOReL, MOPO, COMBO, etc.)

[1] Paine, Tom Le, et al. "Hyperparameter selection for offline reinforcement learning." arXiv preprint arXiv:2007.09055 (2020).

---

> ### Author Response · Authors · 2022-06-05
> **Author Response**
>
> Thank you for the detailed response, it is much appreciated.
>
> We would like to start off by highlighting that there seems to be a misunderstanding about the goals for our problem setting. The problem setting for our work is the online setting, where an agent learns while taking actions in the real-world (in deployment). Given we are in the online setting, we ask: can we use previously collected data to initialize the agent before it has to learn in the real world? Namely, can we use data logs to find suitable hyperparameters for the agent that learns online. We propose to use a calibration model to test many different hyperparameters, and then have to deploy the agent with one hyperparameter setting in the real-world.
>
> This is a common problem setting in industrial control. In fact, we developed this approach to tackle controlling a real water treatment plant. We unfortunately cannot publish the data from those experiments.
>
> We hope this clarifies our setting. Given this, it hopefully helps answer a few of your questions.
>
> **Q1:**
> >“First, I think it is a bit unclear why the authors consider the offline-to-online setting as the whole experiment section.”
>
> The reason for this is that we are not tackling the offline RL setting. We are interested in developing agents that learn online.
>
> You suggest a calibration model could potentially be used to select hyperparameters for offline RL algorithms as well. This indeed might be possible. For example, we could separate the data logs into training data used for the offline RL algorithm and data for the calibration model. We could then evaluate each policy–produced using different hyperparameters and an offline RL algorithm on the training data–by running it in the calibration model.
>
> As mentioned, however, this offline RL setting is not the focus of our work. The algorithm itself is natural–using a calibration model–and so our primary goal is to understand when it might be effective and when it might fail for evaluating agents that learn online.
>
> **Q2:**
> >“Moreover, I think the experiments need to include more comparisons such as [1] and challenging domains such as DM control, DM locomotion, Manipulation playground and tasks from the D4RL benchmark, which will be pivotal to see the effectiveness of the approach.”
>
> We do plan, as next steps, to test in other environments. Here, we wanted to carefully investigate properties of the approach. We picked small environments, but designed several different insightful experiments within these environments. An important first step for an idea is to provide such a careful investigation, before moving on to demonstrations in other environments.
>
> The citation [1] is for offline RL algorithms. As mentioned, we did not find any viable comparators for using a pre-collected batch of data to evaluate an agent that learns online (not a fixed policy).
>
> In light of this potential misunderstanding, we hope that you can update your review and also continue the conversation here. We are more than happy to answer any further questions.

---

### Review · Reviewer_7MV8 · 2022-06-11

**Summary Of Contributions:**

The paper proposes Data2Online, a method of selecting hyperparameters based on a log of offline data. The method is tested empirically on three environments: Acrobot, Puddle world and  Cartpole, providing some evidence of its usefulness.

**Requested Changes:**

I wonder if the authors can provide stronger arguments supporting that their method is likely to be practically useful.

I could imagine various ways. The first one, perhaps the most standard, would be to run more experiments, including more complex environments. There might also be some cheaper ways. Experiments in Section 6.3 suggest that there might be interesting multi-task/CL transfer. Perhaps, one could argue that the calibration data can be reused multiple times (effectively decreasing the cost of their collection). Last but not least, it is possible to pinpoint sample efficiency benefits even with the current data.

**Strengths And Weaknesses:**

Strengths:

- the presented method is simple and elegant
- overall the paper is very well executed
- the paper is well-motivated (i.e. addresses an important problem)
- experiments cover various scenarios

Weaknesses:

- it is not clear if the suggested solution solves the stated problem
- it is unclear how the solution would scale (to more difficult problems)

In more details, the paper proposed *Data2Online*
 method, which, in a nutshell, builds a calibration model used for tuning hyperparameters. The standing assumption on which the method hinges is that the quality of such a model might be relatively weak. I enjoyed reading the paper; the text's quality is high, the structure is well thought out, and, importantly, the experiments cover various situations (e.g., failure case).

Nevertheless, I have doubts about the practical significance. I assume that the ultimate goal is sample efficiency understood in a broad sense, meaning that one takes into account all samples used, including hyperparameter search.

The cost of collecting the offline dataset also needs to be considered (or argued to be irrelevant). Let us consider two extremal cases: near-optimal data, and random exploration data. In the first one, the problem has been solved before (and the prize had been paid). Random exploration data are cheap but have typically poor coverage, therefore making it possible only to find hyperparameters for the initial stages of training. However, perhaps, tuning these initial stages would be cheap to make standard online tuning?

I would imagine that it is (might be) possible to benefit from the proposed method; however, at the moment, I feel the evidence is poor. The environments are somewhat simple (even a few steps suffice to provide good coverage).

A few more details (most of the comments regarding text are nit-picks):

- I find Section 1 and Section 2 slightly too long; for example I’d consider removing some parts of formalization
- I find Section 4.2 a little bit detached from the rest (perhaps some forward references would help?)
- I find Section 4.3 mostly ‘decorative’. Theoretical analysis is able to provide only very pessimistic estimates, which in my view, do not help much in understanding the method. I’d suggest moving it to the appendix
- I like a lot the idea of KNN models.
- I like the solid execution of the experimental section. This softens a little bit my criticism of the choice of environments, i.e. having relatively small and fast ones made it possible to provide a careful statistical evaluation.

---

> ### Author Response · Authors · 2022-06-14
> **Author Response**
>
> We very much appreciate your time in reading our paper and providing us with detailed feedback. We focus on the main concerns below, and will otherwise address the other more minor concerns.
>
> The first concern is about the problem setting. We are motivated by a real application of RL to water treatment (that we are currently working on), so let us use that as an example. The current plant is controlled by an operator and a set of rules. This generates lots of data, under the current control strategy. This data is under reasonable performance (definitely not random), but not optimal either (we can definitely improve on it). This data is essentially free, since it is already being collected. Our main question is: given that there is available data, how do we use it to initialize an online learning agent that is deployed to control the plant and learn online?
>
> This data setting is common in offline RL, and is a reasonable assumption. Many real world problems have such datasets from reasonable policies/decision-makers (often humans). Our problem setting is *not* offline RL; rather our primary goal is online RL—learning and changing the policy in deployment. But, we can similarly leverage datasets that are available.
>
> It is an interesting idea to also consider a setting where the agent can collect some data, build a calibration model, adjust its hyperparameters, and iterate. You are right that in that case we absolutely need to consider the cost of this data gathering. However, here we are not (yet) considering that setting.
>
> The second major concern is about the complexity of the environments. You do mention later in the review that the careful experiments in our simpler environments was appreciated. That is exactly our goal. We designed targeted experiments to test hypotheses, in an environment where we could carefully understand these hypotheses. It is nearly computationally infeasible to run such extensive experiments in popular large-scale RL benchmarks. An important next step is a demonstration in a real environment; we are in fact doing so by using this approach for our water treatment application.

---

### Decision · Action_Editors · 2022-07-17

**Recommendation:** Accept with minor revision

**Comment:**

This work proposes a procedure for tuning the hyperparameters of online reinforcement learning algorithms, using a model estimated using offline interaction data. After describing the proposed approach, the authors show empirical results on three traditional reinforcement-learning testbeds.

The three expert reviewers who assessed this work agree that the proposed approach is simple and intuitive, the topic is relevant, and the paper is clearly written and well executed. On the other hand, they pointed out a series of weaknesses related to the novelty of the proposed approach and its scalability to more complex/real-world domains. Most of these issues have been effectively addressed by the authors in their rebuttal and they have to update their paper accordingly. In particular, since some misunderstandings were common among multiple reviewers, this is a symptom that the writing was not clear enough about those points and the authors need to take particular care in adjusting them.
For what concerns the experimental part, the authors mention that this work was inspired by a real application on a water treatment use case. It would be wonderful if the authors could show results in this domain even by providing them in a qualitative way in order not to violate any non-disclosure agreement.

In conclusion, according to the TML guidelines, we are glad to accept this manuscript, while strongly encouraging the authors to make a final effort to address the reviewers' concerns.

---

> ### Author Response · Authors · 2022-08-02
> **Revised Version Uploaded**
>
> We would like to thank action editors and reviewers for helping us improve the paper. We have now incorporated several of the clarifications requested by the reviewers. We have additionally made the following bigger changes.
>
> 1. In terms of novelty, we have now emphasized more upfront that building and using simulators to prototype algorithms for deployment is a natural idea likely being used in industry. We have particularly emphasized the most similar work mentioned by the reviewer, building simulators for gas turbine control. We have further clarified our particular goal and novelty in this work, and how it contrasts to that work. (We thank the reviewers for helping us see this source of confusion, to better place the work!)
> 2. We have included a discussion (and citations) about using RNNs for the calibration model. We have also modified strong statements about issues with NNs, and rather simply contrasted the KNN approach to that of using NNs, and emphasized that NNs can actually be used to make the KNN approach better (as we do). This is now more clearly separated into its own section, Section 5.4.
> 3. We have emphasized certain aspects of the problem formulation, including (a) that data is collected previously and given to the agent (does not count towards sample use) and (b) why the model needs to be iterated for many steps.
>
> We use our real application as a motivating example throughout, but cannot yet share the results. We are, however, actively working on follow-up, building on the insights we gained from this work.